# RLVR-World: Training World Models with Reinforcement Learning

**Jialong Wu[1], Shaofeng Yin[1,2], Ningya Feng[1], Mingsheng Long[1]✉**
[1]School of Software, BNRist, Tsinghua University    [2]Zhili College, Tsinghua University
wujialong0229@gmail.com, mingsheng@tsinghua.edu.cn

## Abstract

World models predict state transitions in response to actions and are increasingly developed across diverse modalities. However, standard training objectives such as maximum likelihood estimation (MLE) often misalign with task-specific goals of world models, i.e., transition prediction metrics like accuracy or perceptual quality. In this paper, we present RLVR-World, a unified framework that leverages reinforcement learning with verifiable rewards (RLVR) to directly optimize world models for such metrics. Despite formulating world modeling as autoregressive prediction of tokenized sequences, RLVR-World evaluates metrics of decoded predictions as verifiable rewards. We demonstrate substantial performance gains on both language- and video-based world models across domains, including text games, web navigation, and robot manipulation. Our work indicates that, beyond recent advances in reasoning language models, RLVR offers a promising post-training paradigm for enhancing the utility of generative models more broadly. Code, datasets, models, and video samples are available at the project website: https://thuml.github.io/RLVR-World.

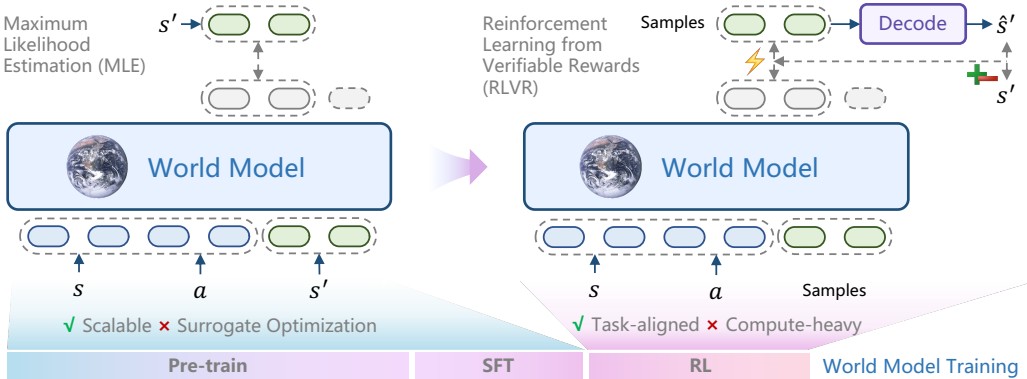

Figure 1: **Training world models with reinforcement learning.** (*Left*) As world models adopt increasingly advanced and scalable architectures, they are typically pre-trained or supervised fine-tuned using surrogate objectives such as maximum likelihood estimation (MLE), which misalign with task-specific prediction metrics. (*Right*) We propose post-training world models via reinforcement learning with verifiable rewards (RLVR) to directly optimize for these metrics.

# 1 Introduction

World models [17, 34], which predict state transitions under action interventions, offer opportunities for autonomously evaluating and optimizing the behavior policies of intelligent agents [20, 21]. By scaling world models across various modalities–such as text [65, 8, 15], video [6, 1], and sensory data [54, 74]–significant progress has been made in a range of applications, including games [7, 30], robotics [73], and autonomous driving [25, 52].

Effectively scaling world models involves training highly expressive models, e.g., Transformers, on massive data. This critically depends on differentiable, rich, and stable training signals. In practice, world models are typically trained using surrogate objectives such as maximum likelihood estimation (MLE). For instance, language models are trained via next-token prediction, which supports reasoning through chain-of-thought generation prior to producing final answers [67], while diffusion models optimize a variational lower bound of the log-likelihood [23]. Moreover, non-end-to-end architectures that incorporate separately trained components like visual tokenizers [11, 51] enhance training efficiency and stability, especially for large-scale models. This paradigm has built powerful probabilistic world models, as well as many other foundation models [4].

However, the ultimate objectives of world models are beyond capturing complex distributions of data, but to meet the usage requirements of transition prediction metrics, such as high accuracy or perceptual quality. Surrogate or non-end-to-end optimizations are often infeasible for, agnostic to, or even diverge from this. In fact, even with differentiable objectives beyond likelihood, non-end-to-end architectures like autoregressive models based on discrete tokenizers or diffusion models cannot directly optimize them. Typical likelihood objectives are not well aligned with the world modeling task: in language models, likelihood-based objectives have been linked to issues like repetition and hallucination [24, 35, 29, 64], and training video models with standard mean squared error is known to produce blurry predictions [43]. Moreover, the widely adopted training paradigm of teacher-forcing next-step prediction is also unaware of accumulation errors over multi-step horizons. A promising emerging approach is tuning pre-trained models to directly optimize toward the target task via reinforcement learning with verifiable rewards (RLVR) [33, 16], which replaces the learned reward model in reinforcement learning from human feedback (RLHF) [48] with a faithful, rule-based reward function. Using this approach, the language model community is now producing impressive advances in tackling complex math reasoning and code generation problems.

In this work, we explore the RLVR paradigm for training world models, referred to as *RLVR-World*. We first propose to unify world modeling across diverse modalities into a general autoregressive generation framework. Concretely, current states and actions are encoded as a sequence of question tokens using modality-specific tokenization schemes, while next states are encoded as response tokens, mirroring the language model formulation. Within this unified framework, we then comprehensively investigate RLVR for world models on two representative modalities:

- **Language world models**: Beyond the success of large language models (LLMs) in math and code domains, we introduce the *world modeling* task as a new testbed for RLVR in LLMs. This task, predicting the transition of verbal world states, naturally lends itself to using prediction accuracy as a verifiable reward. Our experiments show that RLVR can effectively fine-tune LLMs as language world models, yielding significant improvements, including $+\mathbf{30.7}\%$ accuracy on text-based game state prediction [65] and $+\mathbf{15.1}\%$ F1 score on web page state prediction [8].

- **Video world models**: We pioneer the RLVR fine-tuning of autoregressive video world models [69] by directly measuring and optimizing perceptual metrics of decoded predicted frames against ground-truth observations. Notably, our method achieves substantial gains, e.g., $+\mathbf{9.2}\%$ relative improvement on LPIPS [78], on robot manipulation trajectory prediction [5], with merely a few hundred RLVR gradient steps, in contrast to the hundreds of thousands required by MLE training to achieve. It also yields state-of-the-art performance compared with advanced world models [79, 31]. We further prove the effectiveness of RLVR in bridging the gap between pre-trained models and the target world modeling task by showing that it mitigates the repetition issue, a phenomenon commonly seen in LLMs [35] and also observed in our pre-trained video world model.

Finally, we demonstrate the utility of reinforced world models in downstream applications, including policy evaluation [37] and model-predictive control [8]. We hope our method, experiments, and analysis will inspire future research to apply RLVR as a general post-training paradigm to significantly boost the usefulness of world models, and more broadly, generative models.

## 2 Related Work

**World models.**  Learning accurate world models to predict environment state transitions in response to actions is fundamental to model-based planning and reinforcement learning [60]. Due to the inherent complexity and uncertainty of real-world environments, world models are more commonly implemented as generative models of next states conditioned on current states and actions [10, 17], rather than as deterministic models [53, 21]. For visual observations, sequential variational autoencoders have been widely adopted [28, 19, 18, 68], while recent advances have seen the emergence of diffusion-based world models [73, 2, 7], which offer superior visual fidelity. Another important line of work formulates world models autoregressively by discretizing visual inputs into sequences of tokens [45, 25, 69, 1]. This approach naturally extends to modeling additional modalities through unified token-based state representations. For example, previous work explores language-based state representation [65], which has been applied for building world models of the Internet in web agents [8, 15]. Similarly, trajectories of proprioceptive sensors and actuators can also be quantized into token sequences [54, 74]. Motivated by the prospect of building multimodal world models [32, 39] and the potential to leverage the large language model (LLM) ecosystem, our work explores the RLVR paradigm for training world models under the unified autoregressive formulation.

**RL for generative models.**  Reinforcement learning has emerged as a critical paradigm for post-training generative models to better align with human preferences or task-specific objectives. In language models, InstructGPT [48] employs reinforcement learning from human feedback (RLHF) to enhance harmlessness, helpfulness, and honesty. However, RLHF is susceptible to reward model overoptimization [14]. In contrast, DeepSeek-R1 [56, 16] adopts reinforcement learning with verifiable rewards (RLVR), achieving significant advances in math, code, and logical reasoning domains. For visual generative models, text-to-image diffusion models have also been fine-tuned via reinforcement learning [3, 12] to optimize measurable metrics (e.g., compressibility) or human evaluations [70]. Our work identifies world modeling as an underexplored yet natural fit for RLVR, where prediction accuracy serves as a task-aligned, verifiable reward, enabling direct optimization of generative models across various modalities as world models.

## 3 Preliminaries

This section provides a brief background on visual tokenization for unifying visual and verbal state representations, along with the reinforcement learning algorithm used in our work.

**Visual tokenization.**  Given an image $x \in \mathbb{R}^{H \times W \times 3}$, the encoder of a discrete visual tokenizer [63] maps $x$ to its latent representation $h \in \mathbb{R}^{h \times w \times d}$. This latent is then quantized by performing a nearest neighbors lookup in a codebook of embeddings $C = \{e_i\}_{i=1}^K$, yielding a discrete representation $z \in [K]^{h \times w}$, which is passed through a decoder to reconstruct the original image $x$. The token map $z$ can be flattened into a 1D sequence of length $h \times w$ and subsequently modeled by autoregressive models such as decoder-only Transformers [11]. For videos in $\mathbb{R}^{T \times H \times W \times 3}$, a straightforward approach is to tokenize each frame independently using an image tokenizer [45, 7, 39]. However, this leads to excessively long token sequences. To mitigate this, Wu *et al.* [69] propose a compressive tokenization method that exploits temporal redundancy in videos by tokenizing each frame into a reduced number of $n$ tokens $z_t \in [K_1]^n$, conditioned on shared $N$ context tokens $z_c \in [K_2]^N$.

**Group relative policy optimization (GRPO) [56]**  is originally developed for post-training LLMs with reinforcement learning. Compared to PPO [55], GRPO eliminates the need for a value function and estimates advantages in a group-relative manner. Specifically, given a question $q$, GRPO samples a group of responses $\{o_i\}_{i=1}^G$ from the behavior policy $p_{\theta_{old}}$, and computes the advantage of each response by normalizing its reward $R_i$ within the group: $\hat{A}_{i,t} = \frac{R_i - \text{mean}(\{R_i\}_{i=1}^G)}{\text{std}(\{R_i\}_{i=1}^G)}$.

Similar to PPO, GRPO uses a clipped objective with a KL divergence penalty:

$$\mathcal{J}_{\text{GRPO}}(\theta) = \mathbb{E}_{q \sim \mathcal{D}, \{o_i\}_{i=1}^G \sim p_{\theta_{old}}(\cdot|q)}$$

$$\left[ \frac{1}{G} \sum_{i=1}^G \frac{1}{|o_i|} \sum_{t=1}^{|o_i|} \left( \min\left( \frac{p_\theta^{i,t}}{p_{\theta_{old}}^{i,t}} \hat{A}_{i,t}, \ \text{clip}\left( \frac{p_\theta^{i,t}}{p_{\theta_{old}}^{i,t}}, 1-\varepsilon, 1+\varepsilon \right) \hat{A}_{i,t} \right) - \beta D_{\text{KL}}\left[ p_\theta || p_{\text{ref}} \right] \right) \right],$$

$$(1)$$

where $p_\theta^{i,t}$ denotes $p_\theta(o_{i,t} \mid q, o_{i,<t})$ for simplicity. Refer to Shao *et al.* [56] for more details.

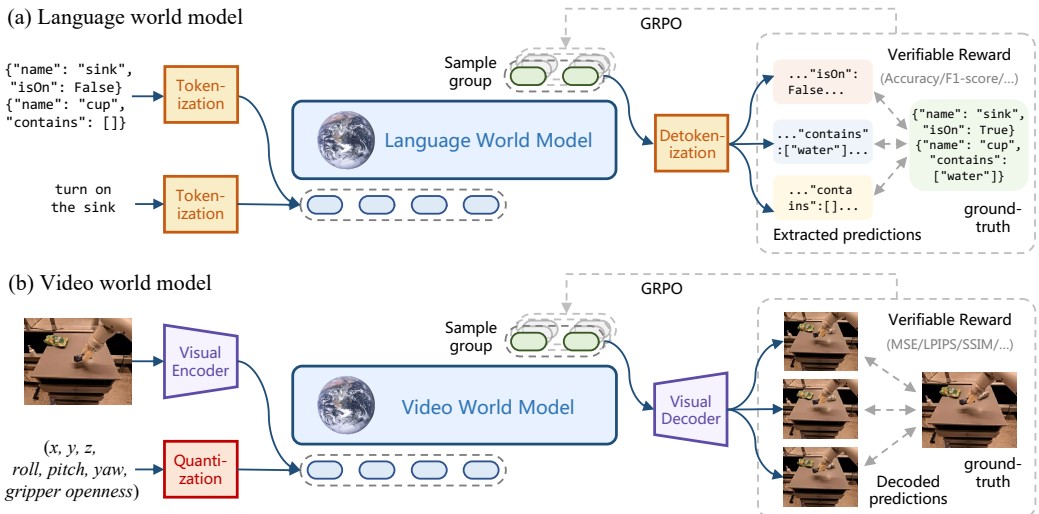

Figure 2: **Illustration of RLVR-World framework.** World models across various modalities are unified under a sequence modeling formulation, and task-specific prediction metrics serve as verifiable rewards. (*Top*) Language-based world models predict verbal state transitions in response to verbal actions. (*Bottom*) Video-based world models, equipped with a visual tokenizer, predict future visual observations conditioned on action vectors.

# 4 RLVR-World: Training World Models with RLVR

As illustrated in Figure 2, this section introduces RLVR-World, a unified framework for training world models across various modalities by reinforcement learning with verifiable rewards.

## 4.1 Problem Formulation

Environments simulated by world models are typically formulated as a Markov decision process (MDP) $\mathcal{M} = (\mathcal{S}, \mathcal{A}, p, r, \gamma)$. Depending on the target task, the state space $\mathcal{S}$ can flexibly include various modalities, such as verbal, visual, or proprioceptive signals. At each timestep, the agent observes a state $s_t \in \mathcal{S}$, takes an action $a_t \in \mathcal{A}$, then transits to a new state according to the distribution $p(s_{t+1} \mid s_t, a_t)$, and receives an immediate reward $r_t \sim r(s_t, a_t)$. More generally, the states can be partially observable and the transitions can be modeled as a $k$-order Markov process $p(s_{t+1} \mid s_{t-k+1:t}, a_{t-k+1:t})$ [1]. World models need to learn to approximate the state transition $p$ and the reward function $r$ accurately. Since rewards can be considered as an extended dimension of the state space [10], our focus is on modeling the transition distribution $p(s_{t+1} \mid s_{t-k+1:t}, a_{t-k+1:t})$.

## 4.2 World Models as Sequence Modeling

While different architectures have been proposed for world models on top of different modalities, next-token prediction by decoder-only Transformers has emerged as a general-purpose formulation applicable to tasks across various modalities. We unify world models into this general sequence modeling framework. To transform states and actions into tokens, different modalities have unique, commonly used tokenization schemes: languages are processed by standard text tokenization techniques like BPE [13]; images and videos are encoded by learned visual tokenizers; and low-dimensional continuous values, e.g., proprioceptive signals, can be quantized to uniform bins over a fixed range.

Then, analogous to language models, we use manually designed templates to construct input token sequences $q(s, a)$ as "questions" and output sequences $o(s')$ as "responses". For simplicity and clarity, here we take the first-order Markov case $p(s' \mid s, a)$ as an example, but the sequence modeling formulation can naturally extend to higher-order cases.

---

[1]While partial *observations* are commonly denoted $o_t$ in classical RL literature, we keep using $s_t$ to avoid confusion with the generated outputs $\{o_i\}_{i=1}^{G}$ in our RLVR context.

We assume an existing world model pre-trained via maximum likelihood estimation (MLE):

$$\mathcal{J}_{\mathrm{MLE}}(\theta) = \log p_\theta(o(s') \mid q(s,a)) = \sum_{t=1}^{|o(s')|} \log p_\theta(o_t(s') \mid q(s,a), o_{<t}(s')). \tag{2}$$

### 4.3 Prediction Metrics as Verifiable Rewards

We then post-train the world model using RLVR to directly optimize verifiable metrics for state transition prediction. Specifically, given an input $q(s,a)$, the pre-trained model generates a group of samples $\{o_i\}_{i=1}^G$, from which the predicted next states $\hat{s}'_i$ are extracted using modality-specific decoding schemes, such as a rule-based extractor for language and a visual decoder for videos. The reward is computed by comparing each prediction in the group to the ground-truth next state $s'$:

$$R_i = \mathrm{sign}(D) \cdot D(\hat{s}'_i, s'), \tag{3}$$

where $\mathrm{sign}(D) = -1$ if lower values of the metric $D$ indicate better predictions (e.g., mean squared error or perceptual loss for visual observations), and $\mathrm{sign}(D) = 1$ otherwise. Using this task-oriented reward, we can fine-tune the world model according to the RL objective in Eq. (1).

**Remarks.** (1) Instead of solving the original environment MDP in Section 4.1, we focus on learning its transition distribution by formulating the next-state prediction process as another MDP, optimized using RLVR. (2) RLVR-World is a general framework, where input/output sequences and reward functions can be domain-specifically designed. We describe them in the experimental sections. (3) Our framework is compatible with various RL algorithms [55, 75], not limited to GRPO.

## 5 Evaluating Language World Models with RLVR

In the following sections, we evaluate the effectiveness of our framework for training world models across different modalities, particularly language (Section 5) and video (Section 6), using RLVR. Inspired by the success in domains like math and code generation, we begin by evaluating our framework on world modeling as a new verifiable task for LLMs, focusing on two domains: text games (Section 5.1) and web navigation (Section 5.2). Experimental details can be found in Appendix A.

### 5.1 Text Game State Prediction

**Dataset and task.** We use ByteSized32-State-Prediction [65], a dataset of text game state transitions for evaluating how well LLMs can serve as text-based world simulators. The dataset contains 76,369 transitions from 31 distinct text games, with 2954 high-quality transitions selected for testing. In this task, an LLM models the world simulator function $F : \mathcal{C} \times \mathcal{S} \times \mathcal{A} \to \mathcal{S} \times \mathcal{R} \times \mathcal{T}$, where $\mathcal{C}$ represents natural language contexts describing the task and action semantics, $\mathcal{S}$ is the state space encoded as JSON objects, $\mathcal{A}$ is the action space, $\mathcal{R}$ is the task reward, and $\mathcal{T} = \{0, 1\}$ indicates task completion.

**World model.** We use DeepSeek-R1-Distill-Qwen-1.5B and 7B [16, 72] as our base model. Due to their limited capability, we first apply supervised fine-tuning (SFT) using responses generated by DeepSeek-R1 [16], and then fine-tune with RLVR using either a binary accuracy reward $R = \mathbb{I}((\hat{s}', \hat{r}, \hat{w}) = (s, r, w))$, or a task-specific one to reflect the problem structure (see Appendix A.1).

**Results.** The results are shown in Table 1. Since predicting state transitions is inherently more difficult when action changes the state, all models perform significantly better on *unchanged* cases. For the 1.5B base model, our RLVR-World with a minimalist binary reward can substantially improve performance over SFT, achieving +34.7% accuracy for *unchanged* and +8.9% for *changed* cases. Incorporating human knowledge through a tailored reward yields even larger gains (+44.8% for *unchanged*, +9.6% for *changed*). When scaled to the 7B base model, our method achieves overall performance surpassing that of GPT-4, although the accuracy on *changed* cases still falls short due to the limited base model's capacity.

### 5.2 Web Page State Prediction

**Dataset and task.** We further evaluate our approach on more realistic web navigation scenarios, using a web page state transition dataset collected by WMA [8] from the WebArena benchmark [80].

Table 1: **Language world model: text game state prediction.** The test set is divided into *unchanged* and *changed* subsets, depending on whether the ground-truth next state differs from the current state.

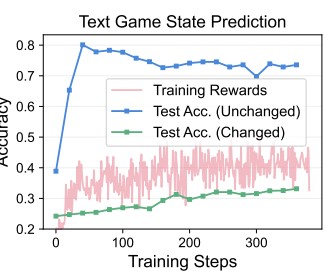

Text Game State Prediction

| Model | Accuracy | | |
| --- | --- | --- | --- |
| | *Unchanged* | *Changed* | *Overall* |
| Base (R1-Distill-Qwen-1.5B) | 11.98% | 0.08% | 7.11% |
| SFT | 38.88% | 24.21% | 32.87% |
| RLVR-World (Ours, binary) | 73.57% | 33.14% | 57.01% |
| RLVR-World (Ours, task-specific) | **83.66%** | 33.80% | 63.24% |
| Base (R1-Distill-Qwen-7B) | 46.90% | 5.53% | 29.92% |
| SFT | 65.94% | 31.32% | 51.76% |
| RLVR-World (Ours, binary) | 83.08% | 40.33% | **65.53%** |
| GPT-4 [65] | 73.90% | **51.60%** | 64.76% |

Table 2: **Language world model: web page state prediction and model predictive control for web agents.** $\Delta$: relative performance gains from RLVR.

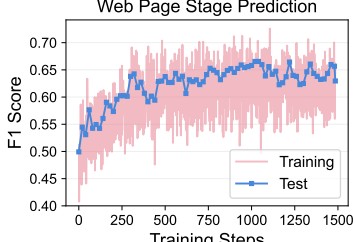

Web Page Stage Prediction

| Model | Web Page State Prediction | | | Web Agent |
| --- | --- | --- | --- | --- |
| | *Precision* | *Recall* | *F1* | *Success Rate* |
| Base (R1-Dist.-Qwen-1.5B) | 15.59% | 15.70% | 11.83% | n/a |
| SFT | 48.99% | 56.05% | 49.94% | 12.06% |
| RLVR-World (Ours) | **72.77%** | **64.55%** | **65.11%** | **14.29%** |
| $\Delta$ | **+48.5%** | **+15.1%** | **+30.3%** | **+18.4%** |

For training and testing, we select a 7K-sample subset consisting of shorter-length samples to avoid out-of-memory issues during training. In this task, a website's state is represented by its accessibility tree, which is simplified from its Document Object Model (DOM) tree, and an LLM is used to predict state transitions after user actions such as clicking. Item changes in the accessibility tree caused by actions are extracted using the Hungarian algorithm, enabling the model to predict these changes directly. Unlike WMA, which generates natural language descriptions of state changes, this design choice facilitates clear verification during the RLVR stage.

**World model.** Following the previous setup, we adopt DeepSeek-R1-Distill-Qwen-1.5B as our base model. We first perform supervised fine-tuning (SFT) using chain-of-thought (CoT) data provided by WMA [8]. Subsequently, we apply RLVR, using the F1 score between predicted item changes $\Delta\hat{s}$ and ground truth $\Delta s$ as the reward function: $R = \mathrm{F1}(\Delta\hat{s}, \Delta s)$. The F1 score, defined as the harmonic mean of precision and recall, is computed by treating precision as the proportion of correctly predicted item changes among all generated ones, and recall as the proportion of correct predictions relative to the ground-truth item changes. An item change is considered correct only if it exactly matches the corresponding ground truth.

**Results.** As shown in Table 2, the world model of the Internet can also be enhanced substantially by RLVR[2], leading to the first key finding of our experiments:

> **Finding on language world models:** *Beyond its success in math and coding, RLVR can also improve LLMs' performance on world modeling tasks involving verbal state transitions.*

### 5.3 Application: Model Predictive Control for Web Agents

Lastly, we show that the reinforced language world models enable more powerful web agents.

**Setup.** Following WMA [8], we build web agents for the WebArena benchmark [80], composed of three components: a policy model, a world model, and a value model. The model predictive control pipeline proceeds as follows: the policy model first proposes multiple candidate actions; the world

---

[2]Due to our novel setup for predicting precise item changes instead of natural language descriptions, direct comparison with established methods like WMA is not feasible. We therefore compare only to our base models.

Table 3: **Video world model: robot manipulation trajectory prediction on RT-1.** We report the mean and standard deviation for each metric calculated over three sampling runs. MSE, LPIPS, and SSIM scores are scaled by 100 for better readability. $\Delta$: relative performance gains from RLVR.

| Task | Model | Repetition Rate↓ | MSE ↓ | PSNR↑ | SSIM↑ | LPIPS↓ |
|------|-------|-----------------|-------|-------|-------|--------|
| *Single-step Prediction* | Base | n/a | $0.336_{\pm0.002}$ | $25.3_{\pm0.03}$ | $81.7_{\pm0.07}$ | $13.0_{\pm0.04}$ |
| | RLVR-World (Ours) | n/a | $\mathbf{0.287}_{\pm0.001}$ | $\mathbf{25.9}_{\pm0.01}$ | $\mathbf{83.1}_{\pm0.00}$ | $\mathbf{12.2}_{\pm0.01}$ |
| | $\Delta$ | n/a | **+14.3%** | **+2.6%** | **+1.6%** | **+6.0%** |
| *Multi-step Prediction* | Base | 48.6% | $0.659_{\pm0.006}$ | $23.1_{\pm0.01}$ | $80.9_{\pm0.03}$ | $14.8_{\pm0.02}$ |
| | Base (w/ repetition rejection) | 0.0% | $0.593_{\pm0.002}$ | $23.3_{\pm0.01}$ | $81.0_{\pm0.02}$ | $14.4_{\pm0.01}$ |
| | RLVR-World (Ours) | 9.9% | $\mathbf{0.486}_{\pm0.003}$ | $\mathbf{24.1}_{\pm0.02}$ | $\mathbf{82.4}_{\pm0.02}$ | $\mathbf{13.4}_{\pm0.02}$ |
| | $\Delta$ | **+79.6%** | **+26.1%** | **+4.5%** | **+1.9%** | **+9.2%** |
| | RLVR-World (w/ rep. penalty reward) | **0.0%** | $0.506_{\pm0.002}$ | $24.0_{\pm0.02}$ | $82.2_{\pm0.01}$ | $13.7_{\pm0.02}$ |

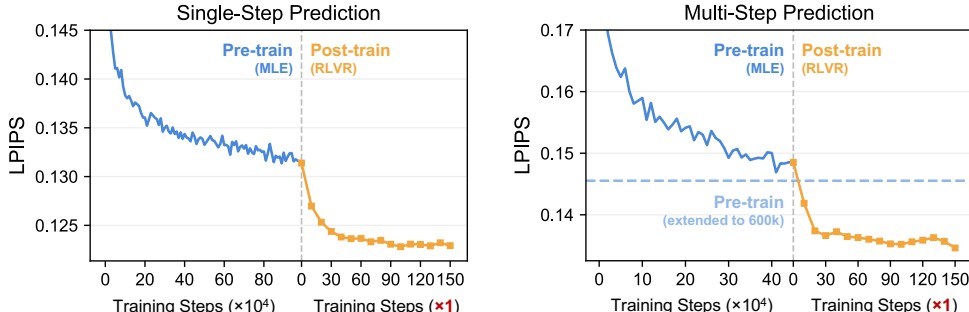

Figure 3: **Learning curves of video world models on RT-1.** Note the significant difference in the $x$-axis scale between the pre-training and post-training stages.

model then predicts the outcomes of these actions; finally, the value model scores each predicted outcome based on the task goal. The action with the highest score is selected for execution. For both the policy and value models, we use DeepSeek-V3 [38], while the world model is taken from our trained models in the previous section. Additional details are provided in Appendix A.3.

**Results.** In Table 2, we compare web agents on top of SFT- and RLVR-trained world models and observe significant improvements. We expect that further gains can be achieved by incorporating stronger policy and value models and extending maximum context length during world model training.

## 6 Evaluating Video World Models with RLVR

We then take a pioneering step in evaluating RLVR to train autoregressive video world models, offering analyses and insights into broader generative models beyond the scope of reasoning models.

### 6.1 Setup

**Dataset and task.** We primarily use the RT-1 robotic manipulation dataset [5] for our experiments, which contains 87,212 tabletop teleoperation trajectories collected from a Google Robot, with 1% left for testing. Each frame of visual observation has a resolution of $256 \times 320$, and the action space consists of 13 dimensions, including arm and base movement. To compare with state-of-the-art models, we also include tabletop pushing (PushT) [9] and deformable object manipulation (Rope and Granular) [77] datasets from DINO-WM [79]. We evaluate world models on two task settings: (1) *Single-step prediction*, formulated as $p(s_{t+1} \mid s_{t-T+1:t}, a_{t-T+1:t})$: predicting the next observation given the past $T$-step observations and actions; (2) *Multi-step prediction*, formulated as $p(s_{t+1:t+T} \mid s_t, a_{t:t+T-1}) = \prod_{i=t+1}^{t+T} p(s_i \mid s_{t:i-1}, a_{t:i-1})$: predicting next $T$-step observation conditioed on the current observation and future action sequence. Predictions are assessed against ground-truth observations using frame-level metrics: MSE, PSNR [27], SSIM [66], and LPIPS [78].

Table 4: **Video world model: comparisons with state-of-the-art on PushT, Rope, and Granular.** We follow the single-step prediction setting and include the baseline results reported in DINO-WM [79]. In addition, we report our reproduced evaluation results using the public DINO-WM checkpoint on PushT. LPIPS and SSIM scores are scaled by 100.

| Model | PushT | | Rope | | Granular | |
|---|---|---|---|---|---|---|
| | LPIPS↓ | SSIM↑ | LPIPS↓ | SSIM↑ | LPIPS↓ | SSIM↑ |
| *Recurrent latent space models* | | | | | | |
| R3M [46, 79] | 4.5 | 95.6 | 2.3 | 98.2 | 8.0 | 91.7 |
| ResNet [22, 79] | 6.3 | 95.0 | 2.5 | 98.0 | 8.0 | 91.5 |
| DINO CLS [47, 79] | 3.9 | 97.3 | 2.9 | 98.0 | 8.6 | 91.2 |
| DINO-WM (Reported) [79] | **0.7** | **98.5** | **0.9** | **98.5** | 3.5 | 94.0 |
| DINO-WM (Public checkpoint) [79] | 3.39 | 96.38 | - | - | - | - |
| *Diffusion models* | | | | | | |
| AVDC [31] | 4.6 | 95.9 | 6.0 | 97.9 | 10.6 | 90.9 |
| *Autoregressive models* | | | | | | |
| Base (Ours) | 0.83 | 98.28 | 3.03 | 97.86 | 3.14 | 94.79 |
| RLVR-World (Ours) | **0.70** | **98.46** | 2.08 | 98.14 | **2.42** | **95.42** |

**World model.** Since no off-the-shelf general-purpose video world models are available, we pre-train variants of iVideoGPT [69] on target datasets as base models by ourselves. For each trajectory segment, observations and actions are tokenized and concatenated into a unified token sequence. We train an image tokenizer to independently tokenize video frames for single-step prediction, but train a compressive tokenizer from iVideoGPT to mitigate sequence length explosion for multi-step prediction. Each action dimension is discretized into 256 uniform bins, determined based on its value range across the entire dataset. During RLVR fine-tuning, we define the reward function as the sum of L1 and perceptual loss between decoded predicted and ground-truth frames: $R = -\sum_{\tau=t+1}^{t+T} [L_1(\hat{s}_\tau, s_\tau) + \text{LPIPS}(\hat{s}_\tau, s_\tau)]$, commonly used in visual tokenizer training [11]. See implementation details in Appendix A.4.

## 6.2 Main Results

As shown in Table 3, RLVR-World significantly improves the base model across all visual metrics on RT-1, demonstrating more accurate and perceptually better video predictions, showcased in Figure 6. We highlight that these gains are achieved with only hundreds of RLVR gradient steps, compared to hundreds of thousands required for MLE pre-training (see the training curves in Figure 3). Even after continuing pre-training for multi-step prediction with 150k additional steps–nearly $1000\times$ more than RLVR fine-tuning–the resulting LPIPS score remains at 14.5, still substantially lagging behind.

**Comparison with state-of-the-art.** We then compare our models against advanced world models, including all recurrent latent state models introduced by DINO-WM [79], which use different latent spaces from pre-trained encoders such as R3M, ResNet, and DINOv2, and an action-conditioned diffusion model, AVDC [31]. As shown in Table 4, after RLVR, our model achieves overall performance comparable to the strongest baseline, DINO-WM, and surpasses it by a significant margin on the most challenging particle-based dataset, Granular. Additional model predictive control results on PushT are provided in Appendix A.4.2.

## 6.3 Model Analysis

**Mitigating repetition.** Prior studies [35] have identified the likelihood objective as a primary cause of repetition in LLM generation. We observe a similar phenomenon in multi-step video prediction, as showcased in Figure 6. This likely stems from the fact that approximately 20% of tokens in each frame remain unchanged from the previous frame, encouraging the model to exploit this shortcut. By directly optimizing video-level prediction metrics rather than next-token likelihood, RLVR effectively mitigates this issue, reducing the repetition rate from 48.6% to 9.9%. To ensure our improvements are not merely due to reducing repetition, in Table 3, we include an additional baseline that repeatedly queries the base model for predictions until a non-repetitive output is sampled.

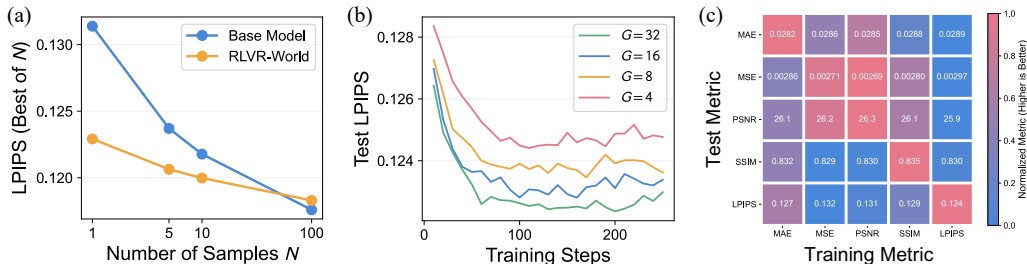

Figure 4: **Model analysis on RT-1.** (a) Test-time scaling: best performance among different numbers of generated samples. (b) RL training scaling: learning curves using different group sizes in GRPO. (c) Metric-oriented optimization: RLVR-World trained and tested on different visual metrics.

**Metric-oriented optimization.** To further demonstrate the effect of direct metric optimization, we post-train five variants of the base model using five different metrics as reward functions: MAE, MSE, PSNR, SSIM, and LPIPS. As shown in Figure 4, models fine-tuned with a specific metric generally achieve the best performance when evaluated on that same metric. Additionally, to show the effectiveness of our method for non-differentiable rewards and achieve zero repetition rate, we introduce an additional repetition penalty reward, defined as the negative rate of consecutive identical frames. In Table 3, after training the model with this extended reward, we observe that it maintains comparable prediction performance while effectively eliminating repetition artifacts.

**Test-time scaling.** We evaluate the test-time scaling behavior of our base and RLVR-trained models by reporting the best metric achieved across $N$ samples [42] in Figure 4. RLVR-World improves one-shot performance, even outperforming the base model's best-of-5 results. This is particularly valuable in practical scenarios where generating large numbers of samples is computationally expensive and ground-truth comparisons are unavailable. However, as $N$ increases to 100, the base model catches up and eventually surpasses RLVR-trained models, echoing findings from Yue *et al.* [76]. This suggests limitations of current RLVR methods and ample opportunities for future research.

**RL training scaling.** While generating more samples at test time is expensive, Figure 4 shows that it is essential for training. Specifically, increasing the group size in GRPO improves both convergence speed and final performance by enhancing sample diversity and expanding the exploration space.

> **Finding on video world models:** *RLVR bridges the gap between pre-training objectives and visual prediction metrics, leading to more accurate predictions, improved training efficiency, and reduced artifacts such as repetition.*

### 6.4 Application: Real2Sim Policy Evaluation

We finally show that our models can serve as real-world simulators for improved policy evaluation.

**Setup.** Following SIMPLER [37], we evaluate four policy checkpoints from RT-1 [5] and RT-1-X [49] on six tasks involving opening and closing top, middle, and bottom drawers. Starting from a real-world observation frame, policies can interact with video world models to roll out neural simu-

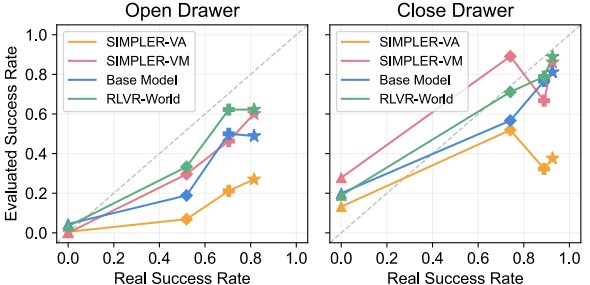

Figure 5: **Real2Sim policy evaluation.**

lated trajectories, allowing for policy evaluation without real-world deployment. Since our preliminary attempts on VLM-based automatic evaluation [61] fail to provide reliable judgments, we rely on human annotators to assess the success of simulated trajectories. Besides our base model, we compare against the simulators developed in SIMPLER. Experimental details can be found in Appendix A.5.

**Results.** As shown in Figure 5, compared to handcrafted SIMPLER simulators, video world models yield smaller discrepancies between real and simulated success rates, suggesting world models as a

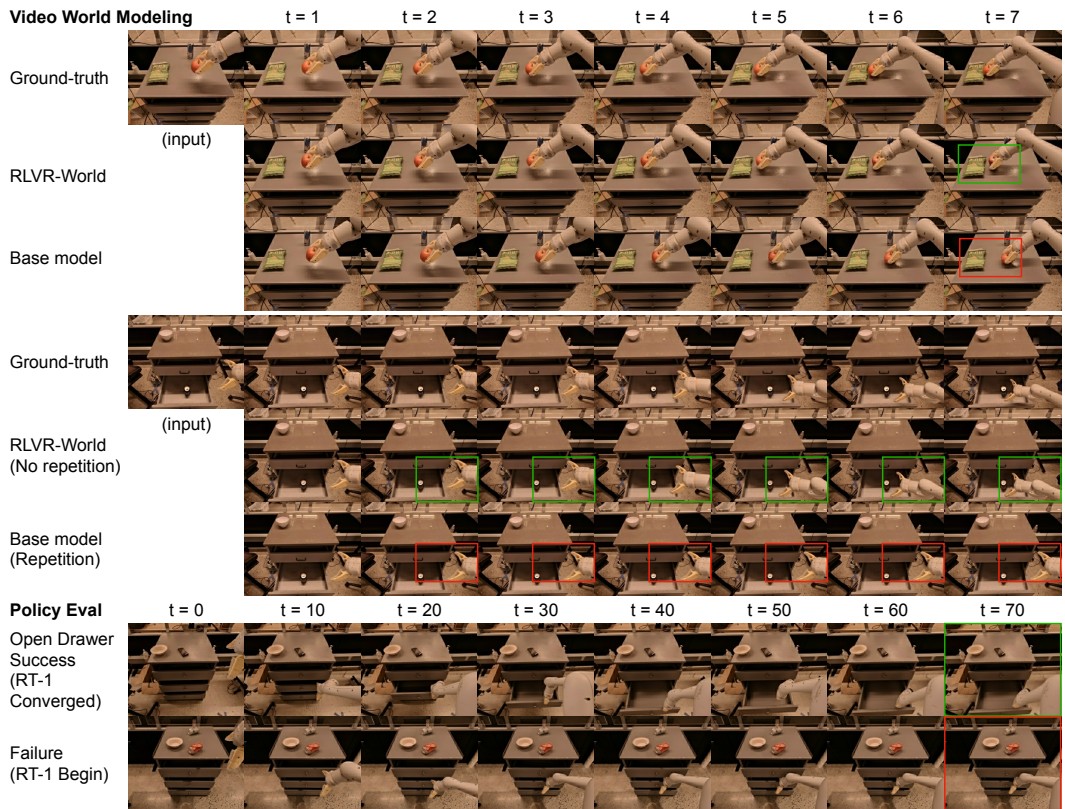

Figure 6: **Qualitative analysis**: multi-step video prediction and policy evaluation on RT-1.

scalable approach to bridging the sim-to-real gap. Among the video world models, RLVR-World further improves upon the base model, achieving more accurate policy evaluation.

> **Finding on model-based applications:** *RLVR-trained world models can improve downstream tasks, including policy evaluation and model predictive control.*

## 7  Discussion and Limitations

In this work, we pioneer RLVR for training world models across language and video modalities, with practical applications in web navigation and robotic manipulation. We believe RLVR has the potential to become a general post-training paradigm for generative models. To this end, several challenges remain for future exploration. **From the algorithmic perspective:** (1) *Breaking performance barriers*: Although RLVR yields significant gains, training typically converges within hundreds of steps. Unlocking continual improvements calls for deeper analysis of bottlenecks in models, data, and algorithms; (2) *Out-of-distribution (OOD) generalization*: Inspired by RLVR's success in enabling LLMs to generalize beyond training domains [57], it is important to study whether similar benefits extend to world models, particularly for counterfactual reasoning on OOD actions in sequential decision-making. **For developing foundation world models:** (1) *Post-training general-purpose world models*: Our current video world models adopt a two-stage training process on the same dataset. We believe that the full potential of RLVR will be further unlocked once the community develops general-purpose video world models [1, 36]. This would enable the complete paradigm of supervised pre-training on diverse domains → supervised fine-tuning → reinforcement fine-tuning, which has already proven effective in our language world model experiments. (2) *Broader application across model classes*: While the core insight behind RLVR-World is model-agnostic, GRPO algorithms for diffusion models [71, 40] are concurrently developed and thus fall outside the scope of our current study. (3) *Task-aligned rewards*: While classical visual metrics align better with the world modeling task than MLE, they still fail to fully capture user-intended qualities. Incorporating constraints such as physical rules and temporal consistency will require more sophisticated reward designs.

## Acknowledgements

We would like to thank many colleagues, in particular Chaoyi Deng and Haixu Wu, for their valuable discussion. This work was supported by the National Natural Science Foundation of China (U2342217 and 62021002), the BNRist Innovation Fund (BNR2024RC01010), and the National Engineering Research Center for Big Data Software.

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

# A Implementation Details and Extended Experiment Results

## A.1 Text Game State Prediction

**Dataset detail.** ByteSized32-State-Prediction [65] comprises 76,369 transitions collected from 31 distinct text-based games. A curated subset of 2,954 high-quality transitions forms the official test set, which we use for both our SFT and RLVR experiments.

The dataset includes task rules written either by human experts or large language models (LLMs). To minimize potential inaccuracies, we adopt the expert-written rules in our experiments.

Each state is represented as a JSON object, describing the environment objects and their associated properties (e.g., a dish with a "clean" or "dirty" status). Actions are described in natural language (e.g., "clean the dish"), and the world simulator aims to predict the resulting state, the task reward based on predefined rules (e.g., +1 for each clean dish), and task completion (e.g., when all dishes are clean). We categorize samples into "unchanged cases" when the action leaves the state unchanged, i.e., the ground truth next state is exactly the same as the current state, and "changed cases" otherwise.

To reduce output complexity, as suggested by the original paper [65], we train our world models for state-difference prediction, generating updates solely for objects whose properties have changed. This design mitigates challenges in generating long, correctly formatted outputs with our 1.5B small base model. Prompt examples are provided in Appendix D.

**Supervised fine-tuning.** To overcome the poor performance of our base model, DeepSeek-R1-Distill-Qwen-1.5B[3], especially its 0.08% accuracy on changed cases (see Table 1), we first apply supervised fine-tuning (SFT) with LoRA [26] to enable effective RL training.

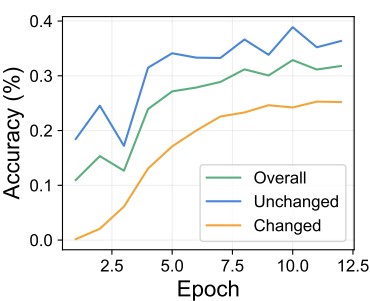

Figure 7: The learning curves of SFT for text game state prediction.

The ByteSized32-State-Prediction dataset lacks suitable SFT data, as it contains only final answers without intermediate chain-of-thought reasoning annotations. To remedy this, we prompt DeepSeek-R1 on a subset of changed cases, apply reject sampling to collect the correct responses, and build a training dataset of 4,237 samples within a $40 budget. We fine-tune the model and report test accuracy across epochs in Figure 7. The checkpoint from epoch 10 is selected as the final SFT result.

**RLVR.** The reward is defined based on the difference between the model's prediction and the ground truth. We explore two reward schemes:

- **Binary reward** assigns a reward of 1 only if the prediction is **completely** correct; otherwise, the reward is 0.
- **Task-specific reward** considers three components: (1) accuracy over all properties $\text{acc}_{\text{all}}$, (2) accuracy over properties that are supposed to change $\text{acc}_{\text{changed}}$, and (3) the binary reward. The final reward is computed as

$$R = \alpha_1 \cdot \text{acc}_{\text{all}} + \alpha_2 \cdot \text{acc}_{\text{changed}} + \alpha_3 \cdot \mathbb{I}(\text{correct}) \tag{4}$$

In our experiments, we use weights $\alpha_1 = 0.1, \alpha_2 = 1, \alpha_3 = 0.2$, which are chosen based on heuristic intuition without further tuning.

To address data imbalance–where over 85% of training samples are unchanged cases–we downsample unchanged cases while retaining all changed cases. For binary reward experiments, we set the changed-to-unchanged ratio at $100 : 40$, and for task-specific reward experiments, we use $100 : 5$, since the task-specific reward more effectively teaches the model to output "no change" when appropriate. Empirically, increasing unchanged samples improves performance on unchanged cases but degrades changed case accuracy, aligning with our expectations. Though this ratio was tuned casually, our current setup sufficiently demonstrates RLVR's effectiveness in improving language-based world models. Hyperparameters for training are provided in Table 5.

---

[3]DeepSeek-R1-Distill-Qwen-1.5B follows the MIT License.

Table 5: Hyperparameters for language world model training.

| | Hyperparameter | Text Game | Web Page |
|---|---|---|---|
| SFT | Batch size | 16 | 8 |
| | LoRA rank [26] | 32 | 32 |
| | LoRA $\alpha$ | 16 | 16 |
| | Epoch | 10 | 4 |
| | Learning rate | $1 \times 10^{-5}$ | $1 \times 10^{-5}$ |
| | Weight decay | 0.01 | 0.01 |
| RLVR | Max response length | 4096 | 7288 |
| | Batch size | 128 | 64 |
| | PPO (GRPO) mini batch size | 64 | 16 |
| | KL loss coefficient | $1 \times 10^{-3}$ | $1 \times 10^{-3}$ |
| | Group size | 5 | 5 |
| | Learning rate | $1 \times 10^{-6}$ | $1 \times 10^{-6}$ |
| Sampling | Top-$p$ | 1.0 | 1.0 for prediction 0.95 for MPC |
| | Temperature | 1.0 | 1.0 |

## A.2 Web Page State Prediction

**Dataset detail.** We use the dataset provided by the WMA repository[4] [8]. As the dataset does not explicitly annotate precise changes to the accessibility tree in response to user actions–i.e., which items are added, removed, or updated–we extract this information using the official script released by the authors[5]. An example of an item is: `[1220] textbox '\ue60c' focused: True required: False`, which includes item ID, type, content, and additional attributes. We can then formulate the next-state prediction task as prompting the LLM to generate all changed items within the accessibility tree. To avoid out-of-memory (OOM) issues during supervised fine-tuning (SFT), we discard samples in which the total length of the prompt and the ground-truth response exceeds 5,000 tokens. After this filtering process, approximately 7,000 trajectories remain from the original 14,000. We allocate 99% of this subset for training and reserve the remaining 1% for testing.

**Supervised fine-tuning.** For our new task formulation, we construct target responses for each sample in the WMA dataset by concatenating the chain-of-thought (CoT) annotations, which are generated by GPT-4o-mini and released by the authors, and our extracted next-state changes. We then perform supervised fine-tuning (SFT) with LoRA [26] on our base model, DeepSeek-R1-Distill-Qwen-1.5B. Training hyperparameters are summarized in Table 5.

**RLVR.** We define the reward function using the F1 score, based on the predicted and ground-truth changed items. Let $\Delta \hat{s} = \{\hat{c}_1, \hat{c}_2, \ldots, \hat{c}_m\}$ represent the set of predicted changed items, and $\Delta s = \{c_1, c_2, \ldots, c_n\}$ the set of ground-truth changed items. We define:

$$\text{True Positives (TP)} = |\Delta \hat{s} \cap \Delta s| = |\{\hat{c} \in \Delta \hat{s} \mid \hat{c} \in \Delta s\}| \tag{5}$$

$$\tag{6}$$

$$\text{Precision (P)} = \begin{cases} \frac{\text{TP}}{|\Delta \hat{s}|}, & \text{if } |\Delta \hat{s}| > 0 \\ 1, & \text{if } |\Delta \hat{s}| = 0 \text{ and } |\Delta s| = 0 \\ 0, & \text{if } |\Delta \hat{s}| = 0 \text{ and } |\Delta s| > 0 \end{cases}$$

---

[4]`https://huggingface.co/datasets/LangAGI-Lab/world_model_for_wa_desc_with_tao_dataset`

[5]`https://github.com/kyle8581/WMA-Agents/blob/main/dataset_construction/scripts/annotation_for_tao.sh`, under the MIT License

$$\text{Recall (R)} = \begin{cases} \frac{\text{TP}}{|\Delta s|}, & \text{if } |\Delta s| > 0 \\ 1, & \text{if } |\Delta \hat{s}| = 0 \text{ and } |\Delta s| = 0 \\ 0, & \text{if } |\Delta \hat{s}| > 0 \text{ and } |\Delta s| = 0 \end{cases}$$

$$\text{F1 Score (Reward } R) = \begin{cases} \frac{2 \cdot \text{P} \cdot \text{R}}{\text{P} + \text{R}}, & \text{if } \text{P} + \text{R} > 0 \\ 0, & \text{otherwise} \end{cases}$$

An item change is considered correct (i.e., a true positive) only if it exactly matches a ground-truth item change across all fields. For example, in the item change [1273] StaticText 'Vortex Running Shoes', the model must produce the exact same string–including all characters and formatting–for the prediction to be considered correct. In cases where no change occurs, the world model is trained to output a special item None, which is treated as a valid item and included in the F1 computation.

RLVR hyperparameters are also listed in Table 5. We select the checkpoint that achieves the highest reward on the test set for final evaluation in the subsequent model predictive control experiments.

### A.3 Model Predictive Control for Web Agents

**Environments.** To ensure consistency with prior work, all experiments are conducted using the official WebArena environment, deployed on an Amazon Web Services (AWS) EC2 instance pre-configured with Docker. WebArena covers five task domains: Shopping, Content Management Systems (CMS), Reddit, GitLab, and Mapping.

**Model predictive control.** Following the method introduced in WMA [8], we use a policy model to sample 20 candidate actions and select the three most frequently sampled for further evaluation. For each selected action, our trained world model performs next-state prediction. A summarization model is then used to (1) identify the top 10 most salient state changes in the predicted next state, and (2) generate a natural language summary describing those changes. This transition summary, along with the task objective, is provided to a value model, which assigns a score between 1 and 5. This scoring query is repeated 20 times, and the final score for each action is computed as the average of these 20 responses. The action with the highest average score is selected for execution. Except for the world model, all models, including policy, summarization, and value, are implemented by prompting DeepSeek-V3 with top-$p$ sampling ($p = 0.95$). All prompts used in model predictive control are detailed in Appendix D.

**Domain-specific results.** We report web agent performance on different domains in Table 6.

Table 6: Domain-specific model predictive control performance for web agents. $\Delta$: relative performance gains from RLVR.

| Methods / Domains | Shopping | CMS | Reddit | Gitlab | Map | Overall |
|---|---|---|---|---|---|---|
| SFT | 18.23% | 11.54% | 4.39% | 6.63% | 19.53% | 12.07% |
| RLVR-World (Ours) | 21.88% | 10.99% | 6.14% | 10.71% | 20.31% | 14.29% |
| $\Delta$ | +20.0% | -4.8% | +40.0% | +61.5% | +4.0% | +18.4% |

### A.4 Robot Manipulation Trajectory Prediction

### A.4.1 RT-1

Hyperparameters of architectures and training process for robot manipulation trajectory prediction are listed in Table 7 and 8.

Table 7: Hyperparameters for visual tokenizers in robot manipulation trajectory prediction. Unless specified, the compressive tokenizer shares the same hyperparameter values with the per-frame tokenizer.

| | Hyperparameter | Value |
|---|---|---|
| Per-frame Tokenizer | Input resolution | $256 \times 320$ |
| | Layers per block | 2 |
| | Channels | $[128, 256, 256, 512, 768]$ |
| | FSQ levels $K$ | $7 \times 5 \times 5 \times 5 \times 5 = 4375$ |
| | Token number $N$ | $16 \times 20 = 320$ |
| | Training steps | $5 \times 10^5$ |
| | Batch size | 16 |
| | Segment length | 8 |
| | Optimizer | AdamW [41] |
| | Learning rate | $5 \times 10^{-4}$ |
| | $L_1$ loss weight | 1.0 |
| | Perceptual loss weight | 1.0 |
| | Adversarial loss weight | 0.1 |
| | Discriminator layers | 6 |
| | Discriminator training start step | 10000 |
| Compressive Tokenizer [69] | Channels | $[128, 256, 256, 512]$ |
| | FSQ levels $K_1$ | $7 \times 5 \times 5 \times 5 \times 5 = 4375$ |
| | Token number $n$ | $8 \times 10 = 80$ |
| | Context FSQ levels $K_2$ | $7 \times 5 \times 5 \times 5 \times 5 = 4375$ |
| | Context token number $N$ | $32 \times 40 = 1280$ |
| | Maximum cross-attention resolution | 32 |
| | Training steps | $6 \times 10^5$ |
| | Batch size | 16 |
| | Segment length | 32 |
| | Number of sampled frames | 7 |

**Dataset.** We use the RT-1 dataset released from Open X-Embodiment [49], which follows the Apache license and contains 87,212 trajectories with $256 \times 320$ visual observations and 13-dimensional actions. Following Wu *et al.* [69], 99% of trajectories are used as the training set and 1% are left as the test set. During testing, a fixed segment from each trajectory is used for prediction.

**Visual tokenizer.** For single-step and multi-step prediction settings, we train a per-frame tokenizer and a compressive tokenizer [69] respectively on RT-1 from scratch. The per-frame tokenizer is essentially a VQGAN [11] with a convolutional encoder-decoder architecture, which tokenizes each frame $s_t$ in the trajectory independently into tokens $z_t \in [K]^N$ where $K$ is the codebook size. The compressive tokenizer is implemented as a conditional VQGAN with dual encoder-decoder pairs $\{(E_c, D_c), (E_p, D_p)\}$. The context encoder $E_c$ first tokenizes a context frame $s_c$ independently into context tokens $z_c \in [K_2]^N$. Subsequently, each frame $s_t$ is tokenized by $E_p$, conditioned on the context encoder's feature maps through cross attention, into tokens $z_t \in [K_1]^n$. Since the context features can capture rich shared information across the trajectory, the number of tokens per frame can be significantly reduced compared to independent tokenization ($n \ll N$). We refer to Wu *et al.* [69] for further details on the compressive tokenizer.

Architectural details of these tokenizers are provided in Table 7. Notably, we adopt finite scalar quantization (FSQ) [44] instead of vector quantization (VQ) [63] within our VQGANs, due to its superior codebook utilization.

For per-frame tokenizer training, we sample batches of trajectory segments and independently reconstruct each frame to compute VQGAN losses. For compressive tokenizer training, the first frames of trajectory segments are used as context frames, and from the remaining frames, we randomly sample a subset to reconstruct for training to reduce memory requirements.

**Sequence modeling formulation.** We consider two task settings for video prediction and describe how token sequences are constructed for autoregressive Transformers:

- **Single-step prediction** $p(s_{t+1} \mid s_{t-3:t}, a_{t-3:t})$: By denoting the tokenized representation of $s_t$ and $a_t$ as $z_t$ and $b_t$ respectively, the token sequence is constructed:

$$x = \text{concat}(z_{t-3}, b_{t-3}, z_{t-2}, b_{t-2}, \cdots, z_t, b_t, [\text{bos}], \underline{z_{t+1}}, [\text{eos}]), \quad (7)$$

where $[\text{bos}]$ and $[\text{eos}]$ are two special tokens[6]. Each dimension of actions is discretized into 256 uniform bins, with all dimensions sharing the same 256 codes. Visual and action codes are offset to avoid overlapping, resulting in a total codebook size of 4633. The final sequence length is $4 \times (320 + 13) + 1 + 320 + 1 = 1654$, where the underlined part of length 321 corresponds to output tokens used for computing the cross-entropy loss.

- **Multi-step prediction** $p(s_{t+1:t+7} \mid s_t, a_{t:t+6}, s_c) = \prod_{i=t+1}^{t+7} p(s_i \mid s_{t:i-1}, a_{t:i-1}, s_c)$: Similarly, the token sequences is constructed as follows[7]:

$$x = \text{concat}(z_c, z_t, b_t, \underline{z_{t+1}}, b_{t+1}, \underline{z_{t+2}}, b_{t+2}, \ldots, \underline{z_{t+7}}, b_{t+7}), \quad (8)$$

where $z_c$ represents the tokenized context frame. Codes for context tokens, per-frame tokens, and actions are offset to avoid index overlapping, resulting in a total codebook size of 9006. The total sequence length is $1280 + 8 \times (80 + 13) = 2024$. Only tokens of frames that need to be predicted contribute to the loss. During generation, predicted frame tokens and action tokens are appended to the sequence alternately.

**Autoregressive transformer.** We adopt the standard architecture of LLaMA [62], instantiated as smaller models with 138M parameters, matching GPT-2 small [50]. Separate Transformers are pre-trained from scratch for single-step and multi-step prediction, respectively. During multi-step prediction training, context frames are sampled from earlier frames preceding the trajectory segment to prevent information leakage for prediction. Architecture and training are detailed in Table 8.

**RLVR.** The pre-trained transformers are fine-tuned with GRPO. The reward function is defined as a negative loss function of all predicted frames and ground truth. Specifically, for single-step prediction:

$$R(\hat{s}_{t+1}, s_{t+1}) = -L_1(\hat{s}_{t+1}, s_{t+1}) - \text{LPIPS}(\hat{s}_{t+1}, s_{t+1}); \quad (9)$$

and for multi-step prediction:

$$R(\hat{s}_{t+1:t+7}, s_{t+1:t+7}) = -\sum_{\tau=t+1}^{t+7} \left[ L_1(\hat{s}_\tau, s_\tau) + \text{LPIPS}(\hat{s}_\tau, s_\tau) \right]. \quad (10)$$

We do not update visual tokenizers during fine-tuning.

**Model analysis.** All experiments presented in Figure 4 are conducted in the single-step prediction setting using a group size of 16, unless specified. Although we find that using a group size of 32 yields slightly better performance, we report performance with a group size of 16 in our main results (Table 3) to maintain consistency with the majority of our experiments.

**Training curves.** We present the curves of training rewards during single-step prediction RLVR training in Figure 8.

**Qualitative analysis.** Additional showcases of video prediction and policy evaluation are presented in Figure 9.

---

[6]We note that these two tokens are not necessary in the sequence formulation but are included due to historical reasons.

[7]$b_{t+7}$ is used here as a placeholder for implementation convenience. Due to the causal architecture and the absence of loss terms on $b_{t+7}$, it completely has no effect on the training process.

Table 8: Hyperparameters for transformers in robot manipulation trajectory prediction. Unless otherwise specified, transformers for single-step and multi-step prediction share the same hyperparameter values.

|  | Hyperparameter | Value |
| --- | --- | --- |
| Architecture | Layers | 12 |
| | Hidden size | 768 |
| | FFN intermediate size | 3072 |
| | Number of attention heads | 12 |
| | RoPE $\theta$ [59] | 10000.0 |
| | Vocabulary size | 4633 for single-step prediction |
| | | 9008 for multi-step prediction |
| Pre-training | Training steps | $9.9 \times 10^5$ for single-step prediction |
| | | $4.5 \times 10^5$ for multi-step prediction |
| | Optimizer | AdamW [41] |
| | Batch size | 32 |
| | Segment length | 5 for single-step prediction |
| | | 8 for multi-step prediction |
| | Learning rate | $5 \times 10^{-5}$ |
| RLVR | Sequence length | 1654 for single-step prediction |
| | | 2024 for multi-step prediction |
| | Optimizer | AdamW [41] |
| | Batch size | 128 |
| | PPO (GRPO) mini batch size | 32 |
| | KL loss coefficient | $1 \times 10^{-3}$ |
| | Group size | 16 |
| | Learning rate | $5 \times 10^{-5}$ |
| | Weight decay | 0.01 |
| Sampling | Top-$k$ | 100 |
| | Temperature | 1.0 |

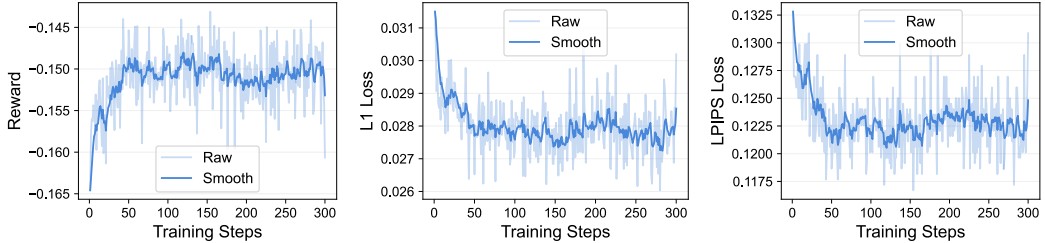

Figure 8: Training curves of RLVR-World for single-step prediction: rewards, $L_1$ losses, and perceptual losses.

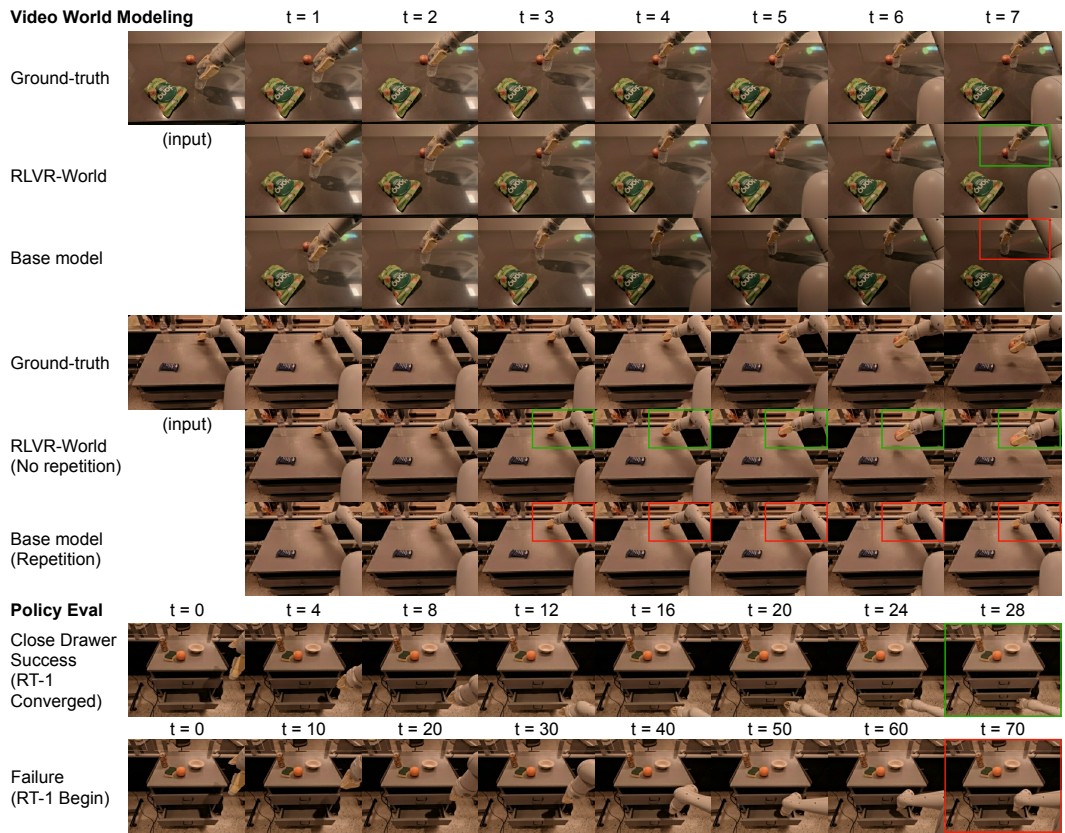

Figure 9: Additional qualitative analysis: multi-step video prediction and policy evaluation on RT-1.

### A.4.2 PushT, Rope and Granular

**Datasets.** We use the datasets provided by DINO-WM [79]. PushT is an environment introduced by Chi *et al.* [9] that features a pusher agent interacting with a T-shaped block. The collected dataset contains 18,685 training trajectories and 21 test trajectories. Each step has a frame resolution of $224 \times 224$ and includes proprioceptive information. The task of the dataset is to predict the next step based on the previous three steps. Rope and granular manipulation tasks are introduced by Zhang *et al.* [77], simulated with Nvidia Flex. Each environment collects 900 training trajectories and 100 testing trajectories, in the resolution of $224 \times 224$. The task of these two datasets is to predict the next step based on the previous one step.

**Implementation details.** Unless otherwise specified, we adopt the same hyperparameters as in Appendix A.4.1 for RT-1. For PushT, proprioceptive information is quantized into uniform bins, consistent with the treatment of action values, and its mean squared error is included in the RLVR reward. For Rope and Granular, given the limited amount of training data, we employ a smaller model with 6 Transformer layers. A group size of 32 is used in GRPO for all three datasets.

**Learning curves.** The learning curves of our models on three datasets are shown in Table 9. Across all datasets, RLVR consistently improves upon our base model. Notably, when training with next-token prediction on smaller datasets (Rope and Granular), the models are prone to overfitting and require early stopping. In contrast, RLVR continues to significantly improve prediction performance on the validation set, demonstrating greater robustness and generalization.

**Extended analysis.** Our model underperforms DINO-WM on the Rope dataset. We hypothesize that this is due to our model being prone to overfitting on this relatively small dataset. In contrast, recurrent latent state models that leverage large-scale pretrained visual encoders demonstrate greater robustness. To validate this hypothesis, we explore training a base model jointly on the Rope and

Table 9: Learning curves of video world models on PushT, Rope, and Granular datasets. LPIPS scores are scaled by 100.

| Training steps | Pre-train | | | RLVR | | | | |
|---|---|---|---|---|---|---|---|---|
| | $1 \times 10^5$ | $2 \times 10^5$ | $3 \times 10^5$ | 60 | 100 | 200 | 300 | 400 |
| PushT LPIPS↓ | 1.19 | 0.90 | 0.83 | 0.78 | 0.75 | 0.74 | 0.72 | **0.70** |

| Training steps | Pre-train | | | | RLVR | | | |
|---|---|---|---|---|---|---|---|---|
| | $2 \times 10^4$ | $4 \times 10^4$ | $6 \times 10^4$ | $7 \times 10^4$ | 100 | 200 | 400 | 720 |
| Rope LPIPS↓ | 3.73 | 3.23 | 3.13 | 3.03 | 2.60 | 2.49 | 2.25 | **2.08** |

| Training steps | Pre-train | | | RLVR | | | | |
|---|---|---|---|---|---|---|---|---|
| | $1 \times 10^4$ | $2 \times 10^4$ | $3 \times 10^4$ | 40 | 100 | 200 | 300 | 500 |
| Granular LPIPS↓ | 3.82 | 3.22 | 3.14 | 2.68 | 2.55 | 2.56 | 2.46 | **2.42** |

Table 10: Performance of video world models jointly trained on deformable object manipulation tasks. LPIPS and SSIM scores are scaled by 100.

| Model | Rope | | Granular | |
|---|---|---|---|---|
| | LPIPS↓ | SSIM↑ | LPIPS↓ | SSIM↑ |
| Base (individually trained) | 3.03 | 97.86 | 3.14 | 94.79 |
| RLVR-World (then individually tuned) | 2.08 | 98.14 | **2.42** | **95.42** |
| Base (jointly trained) | 2.59 | 97.95 | 3.41 | 94.54 |
| RLVR-World (then individually tuned) | **1.65** | **98.29** | 2.44 | 95.40 |

Table 11: Cross-task evaluation of video world models jointly trained on deformable object manipulation tasks. LPIPS and SSIM scores are scaled by 100.

| Model | Rope | | Granular | |
|---|---|---|---|---|
| | LPIPS↓ | SSIM↑ | LPIPS↓ | SSIM↑ |
| Base (jointly trained) | 2.59 | 97.95 | 3.41 | 94.54 |
| RLVR-World (tuned on Rope) | 1.65 | 98.29 | 3.35 | 94.60 |
| RLVR-World (tuned on Granular) | 2.54 | 97.97 | 2.44 | 95.40 |

Granular datasets (both collected from PyFlex simulators), followed by fine-tuning with RLVR on each individual dataset. The results are shown in Table 10. We observe that the LPIPS on Rope improves from 0.0208 to 0.0165. Looking forward, we believe the potential of RLVR can be further unlocked when applied to large-scale pre-trained world models, mirroring the remarkable success of LLMs.

We further investigate whether improvements in one domain can benefit the other. As shown in the Table 11, we observe that when the model is RL-trained exclusively on one dataset, performance on the other can improve slightly. This result aligns with discoveries in the LLM field that improvements in one domain, such as math, can generalize to other domains. However, we also find that performance on the held-out dataset later degrades due to overfitting. We attribute this to the limitations of the base model, which is not pre-trained on large-scale, diverse data and thus lacks truly generalizable representations. We hope these preliminary results can encourage the community to further explore RLVR in the context of large-scale world models.

**Qualitative analysis.** Showcases of our RLVR-World models are presented in Figure 10.

**Model-predictive control.** We strictly follow the same planner configuration as DINO-WM, including goal construction, number of samples, and optimization steps. Since our model predicts discrete tokens and raw pixels without a compact latent space, we use the publicly available DINOv2

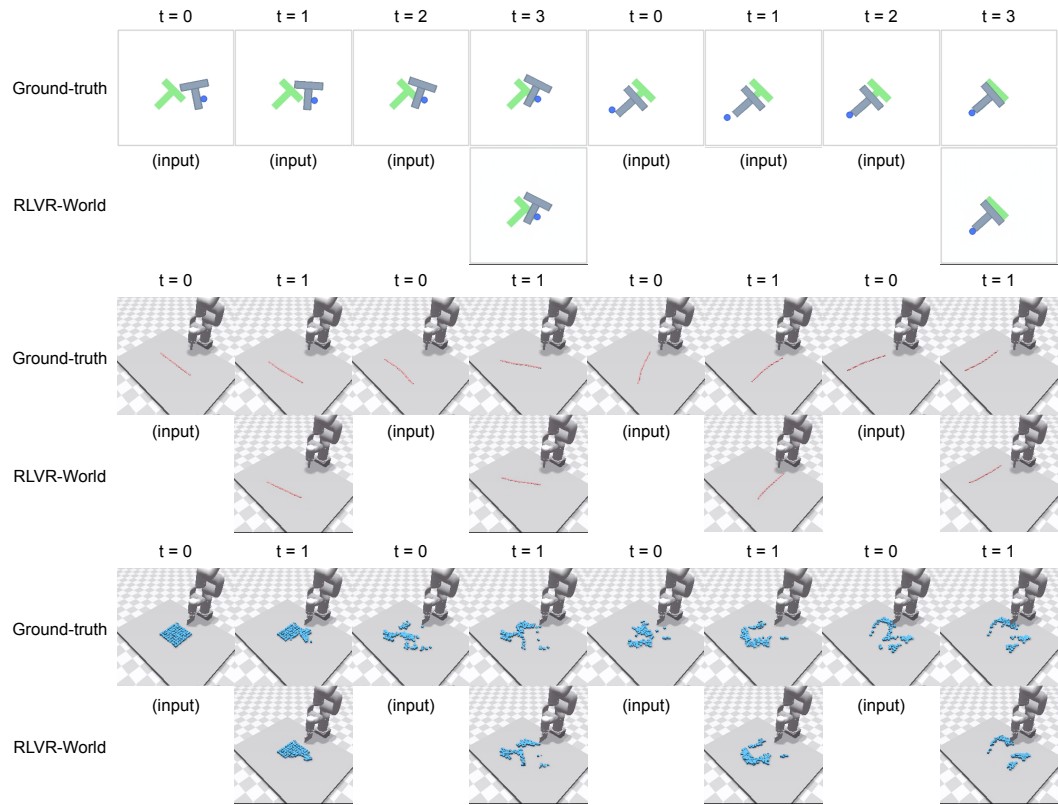

Figure 10: Qualitative analysis: video prediction on PushT, Rope, and Granular.

Table 12: Model-predictive control with different world models on PushT.

| Model | IRIS | DreamerV3 | TD-MPC2 | CEM w/ DINO-WM | CEM w/ Our Base | CEM w/ Our RLVR-World |
|---|---|---|---|---|---|---|
| PushT Success Rate | 0.32 | 0.30 | 0.00 | **0.86** | 0.80 | **0.86** |

encoder to embed predicted frames and compare them to the goal observation during planning. We find that the DINO latent space is more effective for planning than pixel-level MSE or LPIPS, consistent with the findings in DINO-WM. We emphasize that our MPC experiments are intended to compare world models, and the choice of distance metric is orthogonal to our main contributions.

We do not conduct experiments on Rope and Granular due to the lack of MPC configurations and details in the official DINO-WM code, making it difficult to replicate the setup.

The results in Table 12 show that RLVR also enhances the control performance of our base model, achieving results comparable to DINO-WM.

## A.5 Real2Sim Policy Evaluation

**Evaluated policies and baselines.** We aim to evaluate popular open-source generalist policies for the Google Robot, including a series of RT-1[5] checkpoints at different training stages: RT-1 trained to convergence (RT-1 (Converged)), RT-1 at 15% of training steps (RT-1 (15%)), and RT-1 at the beginning of training (RT-1 (Begin)), as well as RT-1-X [49]. SIMPLER [37] is a collection of simulated environments for manipulation policy evaluation on common real robot setups, including different methods to align with real-world environments: "SIMPLER-Visual Matching" (short as SIMPLER-VM in our paper) and "SIMPLER-Variant Aggregation" (short as SIMPLER-VA). We compare our world models against SIMPLER as the representation for policy evaluation methods

based on physical simulation, which requires substantial human effort to develop task-specific environments.

**Task selection.** We experiment with three task types: "pick coke can," "move near," and "open/close drawers," each further divided into subcategories (e.g., "open top/middle/bottom drawer"). In tasks involving the "pick" action, we observe a high rate of false positive evaluations where models incorrectly predict successful trajectories despite poor action quality. This is likely due to the limitation in the expert-level training data, which lacks failed grasp attempts. As a result, the model tends to predict successful grasps even when the robot arm remains some distance from the object. In contrast, the "open/close drawers" task offers more diverse training data, including several failure cases (e.g., drawer slipping from grasp), making it a more suitable evaluation setting for both our base model and RLVR-World.

**Trajectory generation.** For each task, we randomly sample 30 initial frames from the real-world RT-1 dataset to serve as initial observations for world model simulations. Trajectories are then generated iteratively: (1) the policy selects the next action based on the latest frame; (2) the world model predicts the subsequent frame given the action. Following SIMPLER [37], we limit each trajectory to a maximum of 112 frames. Specifically, we use our trained world models for multi-step prediction, able to predict seven future frames conditioned on the current and initial context frame. To predict long trajectories, we apply a sliding window approach, where the last predicted frame becomes the input first frame in the next generation round. To conclude, Our evaluation covers two world models (the base model and RLVR-World), four policies (RT-1 (Begin), RT-1 (15%), RT-1 (Converged), and RT-1-X) [5, 49], six tasks (open/close top/middle/bottom drawer), and 30 trajectories per task, resulting in a total of $2 \times 4 \times 6 \times 30 = 1440$ generated trajectories.

**Success judgment.** We initially explored automatic evaluation of trajectory success using vision-language models (VLMs), including GPT-4o and Gemini 2.0 Flash API. However, both failed to provide reliable evaluations due to two main reasons: (1) results varied significantly with minor changes in the prompt, and (2) the models lack consistent criteria–for example, how much the drawer must be opened to be considered a success varied across scenes and backgrounds. Therefore, we resort to human annotation for evaluation. To ensure consistency and rigor, a single annotator is tasked with labeling success or failure, provided only with the final frame of each trajectory and the task description, without any information about the model or policy used.

**Quantitative results.** The evaluated success rates of different methods are reported in Table 13.

Table 13: Quantitative results for real2sim policy evaluation, corresponding to Figure 5.

| Task | Model | RT-1 (Begin) | RT-1-X | RT-1 (15%) | RT-1 (Converged) |
|---|---|---|---|---|---|
| Open Drawer | Real | 0.0% | 51.9% | 70.4% | 81.5% |
| | SIMPLER-VA | 0.5% | 6.9% | 21.2% | 27.0% |
| | SIMPLER-VM | 0.0% | 29.6% | 46.3% | 60.1% |
| | Base model | 4.4% | 18.8% | 50.0% | 48.9% |
| | RLVR-World | 3.3% | 33.3% | 62.2% | 62.2% |
| Close Drawer | Real | 0.0% | 74.1% | 88.9% | 92.6% |
| | SIMPLER-VA | 13.2% | 51.9% | 32.3% | 37.6% |
| | SIMPLER-VM | 27.8% | 89.1% | 66.7% | 86.1% |
| | Base model | 20.0% | 56.7% | 76.7% | 81.1% |
| | RLVR-World | 18.9% | 71.1% | 78.9% | 88.9% |

# B Computational Resources

**Text game state prediction** The SFT phase uses $4\times$ 80G A100 GPUs for 6.5 hours of training. The RLVR phase is conducted on $8\times$ 80G A100 GPUs over 22.5 hours. Both SFT and RLVR are implemented using the `verl` framework[8] [58].

---

[8]`https://github.com/volcengine/verl`, following the Apache License.

**Web page state prediction**   SFT is performed on $8\times$ 40G A100 GPUs over 17 hours, while the RLVR training is conducted on $8\times$ 80G H100 GPUs for 25 hours. Both SFT and RLVR are implemented using the `verl` framework.

**Robotic manipulation trajectory prediction.**   All experiments in this domain are conducted on a 40G A100 GPU cluster. For single-step prediction, we pre-train the tokenizer using approximately 360 GPU hours. Transformer pre-training takes 530 GPU hours, and RLVR post-training for 200 steps requires 3.5 hours with $4\times$ 40G A100 GPUs. For multi-step prediction, we pre-train the tokenizer with 480 GPU hours. Transformer pre-training takes 500 GPU hours, and RLVR post-training for 200 steps requires 10 hours with $4\times$ 40G A100. Both settings implement tokenizer and transformer pre-training with `accelerate`[9] and RLVR post-training using `verl`.

## C   Broader Impact

**Academic research.**   In the era of foundation models, our work provides a proof of concept that the task performance of generative models can be directly optimized using RLVR, demonstrated through the world modeling task. These results may inspire further research into this paradigm. For instance, task-aligned metrics rather than human preferences could be used to reinforce the visual understanding capabilities of multimodal LLMs in tasks such as visual counting, object detection, and optical character recognition. Additionally, the effectiveness of RLVR as a post-training strategy may further solidify the community's preference for autoregressive generative models over alternatives like masked or diffusion models, as it allows leveraging the well-established LLM ecosystem with minimal modifications.

**Practical applications.**   Enhancing the accuracy of world models through RLVR has clear benefits for real-world autonomous agents, both in digital domains (e.g., web navigation) and physical settings (e.g., robot manipulation), as showcased in our experiments. By enabling agents to simulate the outcomes of actions before execution more accurately, RLVR-trained world models can improve agentic task performance and reduce the risk of harmful behaviors. However, as RLVR is a relatively lightweight post-training stage, its effectiveness is ultimately bounded by the capabilities of the underlying base model. Continued efforts toward building more powerful foundation world models remain essential.

## D   Prompt Examples

In this section, we present detailed prompts used in our LLM experiments to offer a more intuitive understanding of language world models. These prompts are either directly taken from or adapted from prior work [65, 8].

- **Text game state prediction**: We provide examples of the prompts in blue boxes, including the JSON-format game state, rules for action/object/score, as well as the complete prompt. Additionally, we provide an example answer for the state-difference prediction task. They are all kept identical to those in the original dataset.
- **Web page state prediction and web agents** : We present the prompts used in our study in orange boxes. Specifically, we include four types of prompts: *Web State Prediction Prompt Example*, *Top-10 Salient Change Extraction Prompt*, *Next-State Change Summarization Prompt*, and *Web State Evaluation Prompt Example*. The web state prediction prompt and next-change summarization prompt are modified versions of the original prompts proposed by WMA [8], while the top-10 salient change extraction prompt and value prediction (evaluation) prompt are used unchanged, adopted directly from WMA.

---

[9]`https://github.com/huggingface/accelerate`, following the Apache License.

## Text Game: State Difference Prediction Prompt Expample

You are a simulator of a text game. Read the task description and the current environment observation description. Given the current game state in JSON, you need to decide the new game state after taking an action.

Your response should be in the JSON format. It should have three keys: 'modified', 'removed', and 'score'. The 'modified' key stores a list of all the object states that are added or changed after taking the action. Keep it an empty list if no object is added or modified. The 'removed' key stores a list of uuids of the objects that are removed. Keep it an empty list if no object is removed. The 'score' key stores a dictionary with three keys: 'score' is the current game score, 'gameOver' is a boolean of whether the game is over, and 'gameWon' is a boolean of whether the agent won the game. If a player earns a score or wins/loses the game, you should reflect that change in the dictionary saved under the 'score' key. Otherwise, you should set value of the 'score' key to an empty dictionary.Note that while game states can be changed by actions, some game states may change over the time, which is described in the tick function of each object class. Note that while game states can be changed by actions, some game states may change over the time, which is described in the tick function of each object class.

Here are two examples of both cases. Both examples are from the same example game.

Example game task description:

Your task is to wash the dirty dishes.

Here are the descriptions of all game objects properties in the example game:

{OBJECT_RULES}

Here are descriptions of all game actions in the example game:

{ACTION_RULES}

Here is a description of the score function of the example game:

{SCORE_RULES}

In the first example, the game state is changed by an action:

Current observation:

{GAME_OBSERVATION}

Here is the game state:

{GAME_STATE}

The action to take is put dirty plate (ID: 5) in mug (ID: 6)

The expected response is:

{GAME_STATE_DIFFERENCE}

In the second example from the same example game, the game state is changed over the time. Note that while in this example the game state is changed by time only, it is possible that a game state is changed by both an action and time.

Current observation:

{Example_2 observation}

Here is the game state:

{GAME_STATE}

The action to take is eat dishwasher (ID: 2) with dirty plate (ID: 5)

The expected response is:

{GAME_STATE_DIFFERENCE}

Here is the game that you need to simulate:

Task Description:

Your task is to boil water.

Here are the descriptions of all game objects properties:

{OBJECT_RULES}

Here are the descriptions of all game actions:

{ACTION_RULES}

Here is a description of the score function of the game:

{SCORE_RULES}

Current observation:

{GAME_OBSERVATION}

Here is the game state:

{GAME_STATE}

The current game UUID base is 12

The action to take is:

look

**Text Game: Game State Example**

{'game_state': [{'name': 'agent (ID: 0)', 'uuid': 0, 'type': 'Agent', 'properties': {'isContainer': True, 'isMoveable': True, 'isOpenable': False, 'isOpen': True, 'containerPrefix': 'in'}, 'contains': ['plate (ID: 5)', 'mug (ID: 6)', 'knife (ID: 7)']}, {'name': 'plate (ID: 5)', 'uuid': 5, 'type': 'Dish', 'properties': {'isContainer': True, 'isMoveable': True, 'isOpenable': False, 'isOpen': True, 'containerPrefix': 'on', 'dishType': 'plate', 'isDirty': True, 'foodMessName': 'orange'}, 'contains': []}, {'name': 'mug (ID: 6)', 'uuid': 6, 'type': 'Dish', 'properties': {'isContainer': True, 'isMoveable': True, 'isOpenable': False, 'isOpen': True, 'containerPrefix': 'in', 'dishType': 'mug', 'isDirty': True, 'foodMessName': 'sandwhich'}, 'contains': []}, {'name': 'knife (ID: 7)', 'uuid': 7, 'type': 'Dish', 'properties': {'isContainer': True, 'isMoveable': True, 'isOpenable': False, 'isOpen': True, 'containerPrefix': 'in', 'dishType': 'knife', 'isDirty': True, 'foodMessName': 'apple (ID: 11)'}, 'contains': []}, {'name': 'dishwasher (ID: 2)', 'uuid': 2, 'type': 'DishWasher', 'properties': {'isContainer': True, 'isMoveable': False, 'isOpenable': True, 'isOpen': True, 'containerPrefix': 'in', 'isDevice': True, 'isActivatable': True, 'isOn': False, 'cycleStage': 0, 'finishedCycle': False}, 'contains': ['cup (ID: 4)']}, {'name': 'cup (ID: 4)', 'uuid': 4, 'type': 'Dish', 'properties': {'isContainer': True, 'isMoveable': True, 'isOpenable': False, 'isOpen': True, 'containerPrefix': 'in', 'dishType': 'cup', 'isDirty': True, 'foodMessName': 'peanut butter'}, 'contains': []}, {'name': 'bottle of dish soap (ID: 3)', 'uuid': 3, 'type': 'DishSoapBottle', 'properties': {'isContainer': False, 'isMoveable': True, 'isDevice': True, 'isActivatable': True, 'isOn': False}, 'contains': []}, {'name': 'glass (ID: 8)', 'uuid': 8, 'type': 'Dish', 'properties': {'isContainer': True, 'isMoveable': True, 'isOpenable': False, 'isOpen': True, 'containerPrefix': 'in', 'dishType': 'glass', 'isDirty': False}, 'contains': []}, {'name': 'bowl (ID: 9)', 'uuid': 9, 'type': 'Dish', 'properties': {'isContainer': True, 'isMoveable': True, 'isOpenable': False, 'isOpen': True, 'containerPrefix': 'in', 'dishType': 'bowl', 'isDirty': False}, 'contains': []}, {'name': 'banana (ID: 10)', 'uuid': 10, 'type': 'Food', 'properties': {'isContainer': False, 'isMoveable': True, 'isFood': True}, 'contains': []}, {'score': -1, 'gameOver': False, 'gameWon': False}]}

**Text Game: Game State Difference Example**

{'modified': [{'name': 'agent (ID: 0)', 'uuid': 0, 'type': 'Agent', 'properties': {'isContainer': True, 'isMoveable': True, 'isOpenable': False, 'isOpen': True, 'containerPrefix': 'in'}, 'contains': ['mug (ID: 6)', 'knife (ID: 7)']}, {'name': 'mug (ID: 6)', 'uuid': 6, 'type': 'Dish', 'properties': {'isContainer': True, 'isMoveable': True, 'isOpenable': False, 'isOpen': True, 'containerPrefix': 'in', 'dishType': 'mug', 'isDirty': True, 'foodMessName': 'sandwhich'}, 'contains': ['plate (ID: 5)']}], 'removed': [], 'score': {}}

**Text Game: Action Rule Example**

put:
Description: put an object into a target container
Rules:
1. The target must be a container (Container)
2. The target container must be open
3. The object must be in the inventory
4. The object must be moveable (isMoveable)

**Text Game: Object Rule Example**

Object: Container
Description: Abstract class for things that can be considered 'containers' (e.g. a drawer, a box, a table, a shelf, etc.)
Properties:
- A Container is a container.
- A Container could be opened (e.g., e.g. a drawer, a door, a box, etc.), or is it always "open" (e.g. a table, a shelf, etc.).
- A Container has a property indicating if it is opened.
- A Container has a property indicating the prefix to use when referring to the container (e.g. "in the drawer", "on the table", etc.). By default, the prefix is "in"

**Text Game: Score Rule Example**

The player wins the game by getting all dishes clean.
The player gets one point for each dish that is cleaned.
The player loses one point for each dish that is made dirty.

## Web Page: Web State Prediction Prompt Example

You are an intelligent agent that predicts the next state from a given current action using your own logical reasoning. You will be given web-based tasks.

Here's the information you'll have:
- The user's objective: This is the task you're trying to complete.
- The current web page's accessibility tree: A simplified representation of the webpage, providing key information.
- The current web page's URL: This is the page you're currently navigating.
- The previous action: The action you just performed. It may help you track your progress.
- The current action: The action that you will perform next to achieve the user's objective in the current accessibility tree.

The format of previous and current actions can fall into several categories:

**Page Operation Actions:**
`click [id]`: Click an element with a specific id.
`type [id] [content]`: Type the content into the field with the specified id. By default, the "Enter" key is pressed after typing unless `[0]` is appended.
`hover [id]`: Hover over an element.
`press [key_comb]`: Simulate pressing a keyboard combination (e.g., Ctrl+v).
`scroll [down]` or `scroll [up]`: Scroll the page.

**Tab Management Actions:**
`new_tab`: Open a new browser tab.
`tab_focus [tab_index]`: Switch focus to a specific tab.
`close_tab`: Close the currently active tab.

**URL Navigation Actions:**
`goto [url]`: Navigate to a specific URL.
`go_back`: Go to the previous page.
`go_forward`: Go forward (if a previous `go_back` was executed).

**Completion Action:**
`stop [answer]`: Use this when you believe the task is complete. Provide the final answer in brackets if applicable.

To succeed, it's essential to understand how the current action impacts the next state of the webpage. You must verify whether the current action successfully produces the intended effect.

**Reasoning Guidelines:**
1. Generate your answer using logical **REASONING**.
2. **Directly** predict the next state based on the current action.
3. Format your prediction exactly as described below.

**Format Description:** The output is divided into three sections:

- **New items:**
- **Deleted items:**
- **Updated items:**

Each section should be separated by an empty line. If a section has no content, write `None`.

**Web Page: Web State Prediction Prompt Example (continued)**

Each item must follow this format:
`[ID] type 'content' [additional attributes]`

Details:
- `[ID]`: Numeric ID in square brackets.
- `type`: Element type (e.g., `button`, `textbox`, `StaticText`, etc.).
- `'content'`: Element's displayed content in single quotes. Can be empty: `''`.
- `[additional attributes]`: Optional key-value pairs such as `required: False`, `focused: True`, etc.

**Example 1 – New and Updated Items:**
New items:
`[1273] StaticText 'Vortex Running Shoes'`
`[1272] button 'Search'`

Deleted items:
`None`

Updated items:
`[1220] textbox '\ue60c' focused: True required: False`

Now, let's start the task. Here's the task for you:
**URL:** {URL}
**OBJECTIVE:** {OBJECTIVE}
**PREVIOUS ACTION:** {PREVIOUS ACTION}
**CURRENT OBSERVATION:** {CURRENT OBSERVATION}
**CURRENT ACTION:** {CURRENT ACTION}

Please give the result directly in the format of the next state prediction.

---

**Web Agent: Top-10 Salient Change Extraction Prompt**

Summarize the key changes in the web page based on the following information.
New items: {new_items}
Updated items: {updated_items}
Deleted items: {deleted_items}

When summarizing, follow this output format:
1. [First key change]
2. [Second key change]
3. [Third key change]
:
10. [Tenth key change]

## Web Agent: Next-State Change Summarization Prompt

You are an intelligent agent that predicts the next state based on the current action using your own logical reasoning. You will be given a web-based task.

**You will have access to the following information:**
• **The user's objective:** The task you're trying to complete.
• **The current observation:** A simplified representation of the current webpage.
• **The current URL:** The webpage you're currently on.
• **Previous actions:** The action(s) performed just prior to the current one.
• **Current action:** The action being evaluated for its effect on the webpage.
• **Key changes in next state observation:** A summary of differences between the current and next state observations.

**Action types may include:**

*Page Operation Actions:*

- `click [id]`: Click an element with the given ID.
- `type [id] [content]`: Type content into a field with the given ID (append `[0]` to suppress Enter).
- `hover [id]`: Hover over an element.
- `press [key_comb]`: Simulate key combination presses (e.g., `Ctrl+v`).
- `scroll [down]` or `scroll [up]`: Scroll the page.

*Tab Management Actions:*

- `new_tab`, `tab_focus [index]`, `close_tab`

*URL Navigation Actions:*

- `goto [url]`, `go_back`, `go_forward`

*Completion Action:*

- `stop [answer]`: Use this when the task is complete. Provide the answer in the brackets if required.

**Your task: Predict the next state with proper reasoning. Follow these rules:**

1. Begin your answer with: `[Rationale] Let's think step by step...`
2. Your reasoning **must mention** the key changes in the next state observation.
3. Then, **describe the next state** based on those key changes.
4. Output the prediction in this format: `[Next State] The expected effect is that ...`

**Input fields:**

- `URL: {url}`
- `OBJECTIVE: {objective}`
- `PREVIOUS ACTION: {prev_action}`
- `CURRENT OBSERVATION: {cur_observation}`
- `CURRENT ACTION: {cur_action}`
- `KEY CHANGES IN NEXT STATE OBSERVATION: {tao}`

## Web Agent: Web State Evaluation Prompt Example

You are an expert in evaluating and guiding a web navigation agent. Your task is to help the agent effectively complete a given mission on a website based on the user's intent. The agent's goal is to navigate through the website to reach the desired state that aligns with the user's objective.

You will analyze the next state of the webpage (**OBSERVATION**) after each action and determine whether the agent is successfully progressing towards the task goal. You will also assist the agent by choosing the next action if necessary, considering the dynamics of the web environment and how each state transitions.

**Key Points:**

1. **Understand the Intent:**
   - Identify the user's goal (e.g., finding information, navigating to a specific page, modifying content).
   - Ensure the next state of the webpage aligns with achieving that goal based on the current state and user's intent.

2. **Evaluate the Next State:**
   - Assess how the next state contributes to reaching the intended goal.
   - If the next state moves the agent closer to the user's goal, evaluate it positively.
   - If the next state does not progress towards the goal or leads to an error, suggest alternative actions.

3. **State Guidance:**
   - If the agent is on the right track but hasn't completed the task yet, recommend further actions.
   - Guide the agent toward a state that reflects clear progress towards the goal.

4. **Types of Tasks:**
   - *Information Seeking:* The next state must provide specific information (e.g., product price, reviews).
   - *Site Navigation:* The next state must reflect navigation to the exact page or item.
   - *Content Modification:* The next state must show that content has been successfully modified.
   - *General Task:* The final state should reflect that the task is completed. Only issue a `stop` action when the objective is met.

5. **Common Pitfalls:**
   - Avoid corrupted input from repeated typing actions.
   - Ensure the agent navigates to the specific item or content, not just to a general page.

**Output Format with Likert Scale:**
Each next state will be evaluated on a fine-grained scale from 1 to 5:

- **1**: The next state is a failure or leads away from the task.
- **2**: The next state does not contribute meaningfully.
- **3**: The next state is neutral; the agent is maintaining position.
- **4**: The next state is helpful; progress is being made.
- **5**: The next state is optimal; directly aligns with the task goal.

**Response Instructions:**
Write a rationale providing a detailed analysis of the next state and justify your chosen score.
**Output Format:**
`[Rationale]: <your thought> [Score]: <1-5>`

