# OpenReview forum: "RLVR-World: Training World Models with Reinforcement Learning"
_NeurIPS.cc/2025/Conference — NeurIPS 2025 poster_

### Official Review · Reviewer_NuUF · 2025-06-25

**Clarity:** 3
**Significance:** 3
**Originality:** 3
**Rating:** 5
**Confidence:** 3

**Summary:**

World models are becoming increasingly important for robotics, and the current approach of training them through MAE/MSE regression can yield bad results, like blurry outputs. Related works in LLMs have found that RLVR increases performance of sequence modeling, which is related to the objectives of world models. This paper brings RLVR to these world models by defining verifiable rewards in a text exploration environment and using similarity metrics to ground truth video frames in the video world model setup. Results show meaningful improvement of both text and video models after RLVR. World models trained with RLVR have downstream benefits, including planning (MPC) and real-to-sim evaluations.

**Questions:**

Questions

- From Figure 4 Right, it seems like training on LPIPS objective directly can get comparable performance (0.124) to the convergence value of RLVR (Figure 3 Left). Is there an explanation and/or experiment that could differentiate the 0.124 achieved by directly optimizing on LPIPS vs. running RLVR? Or, is the difference significant already?
- Related to the question above, what happens when you train the video world model directly on the reward objective? It could be interesting to see this comparison to gauge the impact of the reinforcement approach over just finding a better objective. If I get these results and there is indeed a significant difference, I’m willing to increase my scores
- The paper mentioned that there are no off-the-shelf video world models, although there is a recent world model (UVA, https://unified-video-action-model.github.io/) that could potentially be relevant. It could be interesting to show results on this or another similar off-the-shelf model to verify your findings, although this is a heavy lift and I don’t expect this to be done during the review process.
- How is repetition defined? In pixel space or in a different representation?

**Ethical Concerns:**

["NO or VERY MINOR ethics concerns only"]

**Final Justification:**

See comment added to the review; In summary, the additional result and clarification removed the main concerns I had about this paper. Although this idea has existed outside of robotics, I believe that this implementation can be a meaningful contribution to the robotics field.

**Limitations:**

Yes

**Paper Formatting Concerns:**

no major concerns

**Quality:**

3

**Strengths And Weaknesses:**

Strengths

- Paper proposes a reasonable application of RLVR to world models: like proofs and science, there are certain facts about the world that are very rule-based, like physics and other causal chains.
- Verified that RLVR works not only on text world models, but also on video world models with meaningful qualitative differences between pre-RLVR and post-RLVR models
- verified that the model can work with MPC in the text environment, showing meaningful downstream improvements (table 2)
- showed that the video model can be used for real-to-sim evaluation and quantiative results show RLVR as closest to corresponding to real success rate.
- good inclusion of repetition rejection as a baseline for multi-step prediction

Weaknesses

- rule-based rewards make sense in the text environment but the connection is weaker for the video model. The objective just becomes another distance metric to a ground truth. A closer analogy to rule-based rewards would be understanding if the laws of physics is being obeyed, if the task is being represented conceptually correctly, etc.
- Related to this past point: GRPO gradient and the objective gradients are different, although on this bandit setting, it’s not immediately obvious what GRPO can do that directly optimizing the reward objective can’t do.
- For video world model, it could be interesting to see other objectives for RLVR to showcase how it can work beyond one type of loss function. In particular, it could be interesting to show how it can work with non-differentiable objectives that can’t be directly optimized upon.
- Summarizing some of the points above: the most notable weakness is with the video model and separating the effects of reinforcement from the choice of objective. I am willing to improve my marks on “significance” and “originality” if a more suitable explanation and/or experiments are done on this point.

---

> ### Author Rebuttal · Authors · 2025-07-31
>
> We sincerely thank Reviewer NuUF for the thoughtful and insightful comments. We hope our response addresses any concerns or misunderstandings and more clearly conveys the value of our work.
>
> ### Q1: Can We Directly Optimize Reward Objectives?
>
> We respectfully suppose that there might be a slight misunderstanding regarding the motivation of our work, which we clarify below:
>
> 1. **Advanced world models (including ours) typically cannot directly optimize task objectives via gradients, due to their non-end-to-end architectures**, as discussed in the Introduction.
>     - Our autoregressive world models consist of a discrete visual tokenizer and a GPT-style transformer. The standard training pipeline involves encoding ground-truth frames into discrete tokens using a pre-trained tokenizer and training the transformer for next-token prediction via a cross-entropy loss. It is not feasible to directly optimize objectives such as pixel MSE or LPIPS by gradients, because **the intermediate steps between transformer and decoder, i.e., sampling predicted tokens and retrieving their corresponding visual embeddings from the tokenizer codebook, are not differentiable**.
>     - Another popular class of diffusion-based world models also face similar problems: they are trained by step-wise denoising objectives, rather than end-to-end target objectives.
> 2. These surrogate training paradigms make **task-specific prediction objectives in video world models well analogous to rule-based reward functions** in LLMs, just as an LLM supervised trained to mimic the chain-of-thought reasoning for math problems does not directly optimize for answer correctness.
> 2. With this insight, our contribution is **not to compete with or enhance direct objective gradient optimization**, but to **enable direct optimization of task objectives in complicated world model architectures where this is previously infeasible**, by leveraging RL. This is why we believe the comparison with direct gradient optimization, as suggested by the reviewer, is not necessary (and actually not feasible) in our context.
> 3. **Clarification on metric-oriented optimization (Figure 4)**: We would like to clarify that in Figure 4, all metrics are optimized using RLVR (instead of end-to-end gradients which is infeasible here), same as our main results. The purpose of this experiment is to show that RLVR can indeed target optimizing desired objectives, similar in spirit to direct gradient optimization in end-to-end architectures. The difference in test LPIPS between training solely for LPIPS (0.124 in Figure 4) and our main result (0.122 in Figure 3, optimized jointly for MAE and LPIPS) is not statistically significant and is expected, given that both setups emphasize LPIPS.
>
> ### Q2: Non-Differentiable Rewards
>
> We indeed agree with the reviewer that performing experiments on non-differentiable rewards, which can not even be optimized by gradients in an end-to-end architecture, will strengthen our experiments, which we conduct as follows.
>
> **Repetition penalty reward (RPR)**: In Section 6.3, we show that RLVR can reduce repetition artifacts in multi-step video prediction (for the formal definition of repetition, see Q4). However, the repetition rate remains non-zero. To address this, we introduce an additional reward term, repetition penalty reward (RPR), defined as the negative rate of consecutive identical frames. We note that this term is non-differentiable. After training the model with this extended reward using RL, we observe that it maintains comparable prediction performance while effectively eliminating repetition artifacts.
>
> | Model                 | Repetition Rate↓ | MSE↓  | PSNR↑ | SSIM↑ | LPIPS↓ |
> | --------------------- | ---------------- | ----- | ----- | ----- | ------ |
> | Base                  | 48.6%            | 0.659 | 23.1  | 80.9  | 14.8   |
> | RLVR-World (w/o RPR)  | 9.9%             | 0.486 | 24.1  | 82.4  | 13.4   |
> | RLVR-World (with PRR) | **0.0%**         | 0.503 | 24.0  | 82.1  | 13.6   |
>
> **On physical law**: We have discussed the use of physical laws as reward functions in the Discussion section. Unfortunately, we currently lack a comprehensive method to detect whether generated videos adhere to physical laws—a challenging problem that still requires further advancements from the research community.
>
> ### Q3: Recent World Model UVA
>
> We greatly appreciate the reviewer bringing up the advanced world model, UVA. While the core insight behind RLVR-World is general and not tied to a specific model class, we note that UVA is implemented as a diffusion model, and its associated GRPO algorithm [2] was developed concurrently with our submission. As such, it falls outside the scope of our current work. We are excited to explore this and will add discussions on applying RLVR for diffusion world models in the next revision of our paper.
>
> ### Q4: Repetition Definition
>
> As our model uses a visual tokenizer to encode pixels into discrete tokens, we define a case of "repetition" when the predicted tokens of the last two frames have exactly the same token IDs, which are also decoded into identical pixel values. Empirically, we find that when the last two predicted frames are the same, it often indicates repetition across many consecutive frames, as showcased in Figure 6.
>
> [1] DINO-WM: World Models on Pre-trained Visual Features enable Zero-shot Planning. ICML 2025.
>
> [2] DanceGRPO: Unleashing GRPO on Visual Generation. arXiv 2025.05.

---

> > ### Comment · Reviewer_NuUF · 2025-08-05
> > **acknowledgement of rebuttal and revision**
> >
> > Hi Authors, thank you so much for the explanation of the misunderstandings and the additional experiments. My main perceived weakness of this paper was with respect to optimizing directly against the objective, and thanks to the content of the rebuttal I see now that there are situations where this would not be directly possible. It would be nice to put this as one of the main motivations in the final publication of the paper. I was reading the evaluations from other reviewers and none of them appeared to mention this explicit advantage either, and it was not quite clear in the introduction of the paper. Motivating with MSE at the beginning (blurry results) was also a bit misleading. However, these are issues of clarity, not method.
> >
> > I appreciate the additional experiment with non-differentiable objective; it is very useful and helps me understand the broader impact of this work. In addition, if this work is indeed the first work to apply RLVR to world models in robotics, I believe that it is a meaningful and publication-worthy contribution to the field, regardless of its incremental improvement with respect to non-robotics work. Keeping these in mind, I will raise my score to an "accept."

---

> > > ### Author Response · Authors · 2025-08-05
> > > **Appreciation for Your Support**
> > >
> > > Dear Reviewer NuUF,
> > >
> > > We sincerely appreciate your strong support and the raised score. We're glad the rebuttal helped clarify our motivation. As you suggested, we will revise the paper to more accurately reflect the core insight of our method, which we believe will substantially improve our paper.
> > >
> > > Best regards,
> > >
> > > Authors

---

### Official Review · Reviewer_d8tC · 2025-06-26

**Clarity:** 4
**Significance:** 2
**Originality:** 3
**Rating:** 5
**Confidence:** 4

**Summary:**

The authors introduce RLVR-World, a novel unified framework that leverages reinforcement learning with verifiable rewards to directly optimize world models for downstream task objectives. Through extensive evaluations on both language- and video-based benchmarks, RLVR-World achieves significant and consistent performance improvements across diverse domains.

**Questions:**

- The scope of the evaluation is currently limited to narrowly-defined, task-specific scenarios (e.g., RT1). By contrast, DeepSeek-R1 applies the RLVR framework across a diverse set of domains—math, code, reasoning—and demonstrates consistent gains, suggesting that RLVR can endow models with broader “general” capabilities. Could the authors extend their analysis to assess whether RLVR likewise enhances the general-purpose predictive power of the world model? Without such analyses, the reported results remain consistent with the well-known benefits of the pretrain–finetune paradigm rather than demonstrating novel, task-agnostic advances.
- About Figure 5, a smaller measured discrepancy does not necessarily correspond to a reduced sim-to-real gap. For example, imagine evaluating 100 trajectories: the first 50 succeed only in the real environment and the remaining 50 succeed only in simulation. In this case, the discrepancy metric would be zero, yet the sim-to-real performance gap is maximal.

**Minor Comments**

- (Line 220) “observation conditioed on” -> “observation conditioned on”

**Ethical Concerns:**

["NO or VERY MINOR ethics concerns only"]

**Final Justification:**

The authors propose RLVR, an effective method for directly optimizing task objectives in world models. While RLVR has demonstrated success in the context of large language models, its successful adaptation to world modeling—along with comprehensive empirical validation—represents a meaningful contribution.

**Limitations:**

The authors discuss the limitations of their work in the paper.

**Paper Formatting Concerns:**

No paper formatting concerns

**Quality:**

3

**Strengths And Weaknesses:**

**Strengths:**

- Extends the RLVR framework—originally formulated for pure LLM settings—to vision-based world models, thereby broadening its applicability.
- The authors conduct extensive experiments on both language- and video-based world models across multiple domains, demonstrating the robustness and generality of their approach.
- Overall, the paper is well written and easy to follow.

**Weaknesses:**

- The paper’s central paradigm—leveraging a pretrained model followed by task-specific fine-tuning (whether via supervised learning or reinforcement learning)—is by now a well-worn recipe, particularly in NLP. Extending this two-stage approach to autoregressive video world models is sensible, but it does not in itself constitute a surprising or fundamentally new insight, especially a number of prior works have already framed world-model learning as autoregressive sequence prediction over discretized multimodal tokens as mentioned in section 2.

---

> ### Author Rebuttal · Authors · 2025-07-31
>
> We would like to sincerely appreciate Reviewer d8tC for the comprehensive review and insightful questions.
>
> ### Q1: Clarification on Our Insight and Contribution
>
> We would like to clarify that our central paradigm contribution is **not simply performing task-specific fine-tuning after pre-training**, but rather **revealing RLVR as a more task-aligned fine-tuning paradigm for world models**, compared to conventional supervised fine-tuning (referred to as maximum likelihood estimation, or MLE, in our paper). In other words, our key insight addresses *how* fine-tuning should be done, rather than *whether* fine-tuning should be done at all.
>
> As we discussed in the Introduction, advanced world models, whether autoregressive or diffusion-based, often employ non-end-to-end architecture and surrogate training objectives. While such approaches are scalable and well-suited for pre-training, we point out that **continuing to use the same training methods for task-specific fine-tuning is not effective enough**. This is because these surrogate training methods misalign with the actual task objectives of world models. Our insight is that switching to a more aligned training paradigm, RLVR, can lead to significantly faster and more effective post-training. For example, as shown in Figure 3, **training speed of RLVR can be more than 1000x faster**. Such results are **not already well-known**, sufficiently surprising, and worth sharing with the community.
>
> We acknowledge that the current form of Figure 1 may have caused confusion regarding our paradigm contribution. We will revise it in the next version to more clearly emphasize the differences between supervised and RLVR fine-tuning.
>
> ### Q2: Enhancing General-Purpose Capability
>
> The mechanism by which RLVR improves general capabilities in LLMs such as DeepSeek-R1 remains an open and active research question. A common belief is that the base model must be sufficiently strong for task-specific improvements (e.g., in math or code) to transfer to broader domains. Unfortunately, training LLMs with hundreds of billions of parameters exceeds our available computational resources. Moreover, at present, no video world model exists with similarly strong general capabilities as LLMs. For this reason, we focus our experiments on a specific scenario (RT-1). Nevertheless, we provide additional experiments below using a "general" video world model across two relevant domains.
>
> **Preliminary experiments**: We adopt two datasets, Rope and Granular,  from DINO-WM [1], a related work suggested by Reviewer BbTo. Both are collected from the same simulator, PyFlex, but involve different tasks. A shared base model is jointly trained on the two datasets, and then fine-tuned by RLVR on each dataset individually, to investigate whether improvements in one domain can benefit the other.
>
> | Method                   | LPIPS ↓ | LPIPS ↓  | SSIM ↑ | SSIM ↑   |
> | ------------------------ | ------- | -------- | ------ | -------- |
> |                          | Rope    | Granular | Rope   | Granular |
> | Base (jointly trained)   | 0.0259  | 0.0341   | 0.9795 | 0.9454   |
> | RLVR-World (on Rope)     | 0.0165  | 0.0335   | 0.9829 | 0.9460   |
> | RLVR-World (on Granular) | 0.0254  | 0.0244   | 0.9797 | 0.9540   |
>
> As shown in the table above, we observe that when the model is RL-trained exclusively on one dataset, performance on the other can improve slightly. This result aligns with discoveries in the LLM field. However, we also find that performance on the held-out dataset later degrades due to overfitting. We attribute this to the limitations of the base model, which is not pre-trained on large-scale, diverse data and thus lacks truly generalizable representations. We hope these preliminary results can encourage the community to further explore RLVR in the context of large-scale world models.
>
> ### Q3: Discussion on Policy Evaluation Metrics
>
> The goal of real2sim policy evaluation is to **identify effective policies using only rollouts from simulated environments**. In this setting, our focus is not on reducing the "sim-to-real gap", a topic already addressed in detail in our prediction experiments (Sections 6.2 and 6.3). Instead, we aim to assess whether the model can accurately estimate the success rate of a given policy. In this context, a smaller measured discrepancy is sufficient to demonstrate the model’s effectiveness.
>
> This formulation aligns with the standard setup in most existing policy evaluation studies. For example, [2] introduced a benchmark using metrics like rank correlation to compare policy rankings derived from success rates in simulators and the real world. Similarly, [3] and [4] employed metrics like Pearson correlation and regret@1, both of which are computed over aggregated success rates rather than individual trajectories. SIMPLER [5], a collection of simulated environments on common real robot setups, also takes its accurately estimated success rates as evidence to demonstrate that simulation-based evaluation can be a scalable, reproducible, and reliable proxy for real-world evaluation.
>
> [1] DINO-WM: World Models on Pre-trained Visual Features enable Zero-shot Planning. ICML 2025.
>
> [2] Benchmarks for Deep Off-Policy Evaluation. ICLR 2021.
>
> [3] Autoregressive Dynamics Models for Offline Policy Evaluation and Optimization. ICLR 2021.
>
> [4] Variational Latent Branching Model for Off-Policy Evaluation. ICLR 2023.
>
> [5] Evaluating Real-World Robot Manipulation Policies in Simulation. CoRL, 2024.

---

> > ### Comment · Reviewer_d8tC · 2025-08-05
> >
> > Thank you to the authors for their thoughtful responses and additional results presented during the rebuttal phase.
> >
> > The experiments regarding the general-purpose capabilities of the approach, along with the explanation of the policy evaluation metrics, adequately addresses my questions.
> >
> > However, I remain unconvinced about the novelty of the core contribution. The authors argue that their central contribution is “revealing RLVR as a more task-aligned fine-tuning paradigm for world models”. I do agree with this point, but the close relationship between reinforcement learning and world models is already well established as noted in the paper’s Related Work, “Learning accurate world models to predict environment state transitions in response to actions is fundamental to model-based planning and reinforcement learning”. Consequently, applying RL-based fine-tuning rather than supervised fine-tuning may be seen by many(at least myself) as a natural and expected choice.
> >
> > I appreciate the authors’ efforts and the quality of the work, but my decision is to maintain my score.

---

> > > ### Author Response · Authors · 2025-08-05
> > > **Thanks for the response and further clarification**
> > >
> > > Dear Reviewer d8tC,
> > >
> > > We sincerely appreciate your response and fully respect your decision to maintain the score. However, we would like to offer a briefly extended clarification regarding the positioning of our contribution.
> > >
> > > When we stated in the Related Work section that "*learning accurate world models to predict environment state transitions in response to actions is fundamental to model-based planning and reinforcement learning*," we were referring to the well-established paradigm of using world models for **reinforcement learning policies (WM for RL)**. The typical pipeline of model-based RL is as follows:
> > >
> > > > Collect data from the real environment → train a world model on this data using **supervised learning** → train a policy within world model simulations using **reinforcement learning**
> > >
> > > In contrast, our work proposes to **reinforcement learning world models themselves (RL for WM)**. The above pipeline then becomes:
> > >
> > > > Collect data from the real environment → train a world model on this data using **reinforcement learning** (optionally after supervised pretraining) → train a policy within world model simulations using **reinforcement learning**
> > >
> > > To the best of our knowledge, this is the first work to explore and formalize RL-based world model training. It introduces a new paradigm that complements the previously one-direction "close relationship between reinforcement learning and world models" (WM for RL), by demonstrating the benefits of the opposite (RL for WM).
> > >
> > > We are encouraged that our contribution has been recognized by Reviewer NuUF, who described our first application of RLVR to world models as "a meaningful and publication-worthy contribution to the field."
> > >
> > > Once again, thank you for your time, effort, and constructive feedback throughout the review process!
> > >
> > > Best,
> > >
> > > Authors.

---

> > > > ### Comment · Reviewer_d8tC · 2025-08-06
> > > >
> > > > Thank you for the authors’ thoughtful response. I acknowledge the paper as a pioneering effort in formulating and exploring RL-based training for world models—an important conceptual contribution. However, from this perspective, a key claim of the work is that RLVR provides a **more** task-aligned fine-tuning paradigm compared to SFT, and this claim requires stronger empirical support.
> > > >
> > > > In Section 5 (Language World Models), the use of non-differentiable rewards (e.g., task-specific rewards, F1 score) indeed necessitates reinforcement learning, as SFT cannot directly optimize such objectives. However, since SFT is applied only as a pre-RL step due to the base model’s limitations, the current setup does not allow for a direct comparison between SFT and RLVR on the same task. Consequently, the results do not establish RLVR as superior to SFT in aligning models with task objectives.
> > > >
> > > > In Section 6 (Video World Models), task objectives like MSE are differentiable but SFT, as a viable alternative, is not compared. The authors highlight RLVR’s faster training speed—claiming over 1000× speedup in rebuttal. Yet, this comparison appears unfair: pretraining on multiple datasets (SFT) versus fine-tuning on a single task (RLVR) involves fundamentally different training paradigm. Such speed advantages do not necessarily reflect the efficiency or effectiveness of RLVR as a fine-tuning method.
> > > >
> > > > To strengthen the paper’s central claim, I would encourage the authors to provide a controlled comparison between SFT and RLVR. Demonstrating either superior performance or higher sample efficiency of RLVR over SFT in such a setting would significantly bolster the paper’s impact.
> > > >
> > > > I am open to raising my score if the authors can include such evidence or clarify these comparisons.

---

> ### Author Response · Authors · 2025-08-06
> **Clarification on Comparison Between SFT and RLVR**
>
> Dear Reviewer d8tC,
>
> We sincerely appreciate your thoughtful response and engagement in such a meaningful discussion. We would like to offer a detailed clarification regarding the comparison you requested.
>
> First of all, we apologize for any confusion caused by our inaccurate wording. When we use the term "task-aligned," we do not mean alignment to a specific **domain** of data (e.g., pre-training data vs. downstream data), but rather alignment to a specific **objective** (e.g., conventional cross-entropy vs. task objectives like LPIPS) within the same domain.
>
> We can now discuss the advantages of RLVR over SFT, both principally and empirically.
>
> **Principally**:
>
> - As the reviewer correctly noted, many task objectives (e.g., F1 score) are non-differentiable and thus cannot be optimized directly. SFT can only optimize the cross-entropy loss of the desired outputs, serving as a **surrogate training objective**.
> - Even for differentiable task objectives (e.g,. MSE or LPIPS), direct optimization is often infeasible due to non-end-to-end architectures commonly used in modern advanced world models. We can only rely on surrogate optimization, too. For example, our transformer-based video world model first tokenizes ground-truth frames and then learns via next-token prediction. For a more in-depth discussion in this aspect, please see our response for Reviewer NuUF (*Q1: Can We Directly Optimize Reward Objectives?*).
> - In contrast, RLVR overcomes limitations by either non-differentiable objectives or architectures. It **directly optimizes desired task objectives** via policy gradients. This is what we refer to as being more task-aligned.
> - A more precise statement of our contribution might be identifying an effective method (RLVR) to **enable such direct optimization of task objectives in world models where this is previously infeasible**. This contribution has been well recognized by Reviewer NuUF, who subsequently raised the score to 5 ("accept").
>
> **Empirically**:
>
> - In the context of our discussion, the comparison of SFT and RLVR then becomes a comparison between surrogate and direct optimizations, using the same downstream data. Across all experiments of our paper, this has been done by **comparing how much performance gain can be achieved from the same base checkpoint using either SFT or RLVR**.
> - In the **language world model** experiments, SFT indeed is applied as a pre-RL step to get a checkpoint with basic task performance. However, let's consider a thought experiment: if we **continue SFT** on this checkpoint, task performance would stagnate or even overfit (since we already apply early stopping to get the SFT checkpoint in our experiments). However, by **switching to RLVR**, our experiments show that we can achieve further performance gains. We believe this is already a controlled comparison, which can illustrate the superiority of RLVR over SFT in "aligning models with task objectives."
> - In the **video world model** experiments, the same comparison is also explicitly presented in the original submission. We pre-train our video world model (solely) on the RT-1 dataset with $10^5\sim 10^6$ steps to obtain a base checkpoint (with LPIPS 14.8). **Continuing to train** on the same dataset with the same standard cross-entropy loss (which is analogous to SFT) yields only slow improvements: As mentioned in Line 235, 150k additional steps can only achieve an **LPIPS 14.5**. In contrast, **switching to RLVR** from the base checkpoint (see Figure 3) quickly achieves a significantly better **LPIPS 13.4**, with just 100 gradient steps (a more than 1000x speedup). This comparison **from the same base checkpoint** also sufficiently "demonstrates either superior performance or higher sample efficiency of RLVR."
>     - **[!!! Important]** We would like to respectfully correct **a factual misunderstanding in the reviewer's comment**. Since the community currently does not have a strong enough general-purpose world model, we restrict our entire training pipeline (including pre-training, continual pre-training, and RLVR) **solely on the RT-1 dataset**. We sincerely apologize for not explicitly stating this in the paper (only Lines 564 and 599 in the appendix contain vague descriptions). Thus, we did **NOT** compare "pretraining on multiple datasets (SFT) versus fine-tuning on a single task (RLVR)."
>
> We will carefully revise our paper to more clearly present the above comparison. We hope our detailed clarification helps address your concerns and strengthens your evaluation of the value of our work.
>
> Best,
>
> Authors.

---

> > ### Comment · Reviewer_d8tC · 2025-08-06
> >
> > Thanks to the authors for their clarification. I will increased my score to 5.
> >
> > Regarding the video world model experiments, I appreciate the clarification that both pretraining and fine-tuning are conducted on the same dataset (RT-1). Thus this experiment appears to be more like a two-phase training rather than pretrain-finetune paradigms with domain shift. I would encourage discussion about whether such a two-phase training strategy has broader applicability or particularly suitable for world modeling. A brief analysis would strengthen the theme “Training World Models with Reinforcement Learning”.

---

> > > ### Author Response · Authors · 2025-08-06
> > > **Deep Appreciation for the Constructive Discussion**
> > >
> > > Reviewer d8tC,
> > >
> > > We sincerely thank you for your time and effort in engaging in such an in-depth and constructive discussion, which has helped us resolve all remaining concerns.
> > >
> > > In response to your latest feedback, we would like to offer a few remarks:
> > >
> > > - We agree that our video world model experiment adopts a two-stage training process, **supervised learning → reinforcement learning**, on the same dataset. But we believe that the full potential of our insight and methodology will be further unlocked in the future when the community develops general-purpose video world models. This would enable the complete paradigm of **supervised pre-training on diverse domains → supervised fine-tuning → reinforcement fine-tuning**, which has already proven effective in our language world model experiments. We will clarify the current training setup of our video world model and expand the discussion of this future direction in the revised paper.
> > > - We also believe the methodology, using RLVR as a more task-aligned training paradigm to SFT, is broadly applicable and is gaining growing consensus in the broader AI community. It has already achieved impressive success in domains like math and code with LLMs. Our particular contribution lies in identifying its suitability for world modeling, a rapidly evolving field that faces similar challenges due to increasingly complex architectures. We view this as a paradigm-shifting insight for the field, as demonstrated by our successful application of RLVR to world models across both language and video modalities. We will highlight this domain-specific insight in the revision.
> > >
> > > Thank you again for your support and for recommending our work for acceptance.
> > >
> > > Best,
> > >
> > > Authors.

---

### Official Review · Reviewer_ndrJ · 2025-07-02

**Clarity:** 2
**Significance:** 3
**Originality:** 3
**Rating:** 4
**Confidence:** 2

**Summary:**

This paper presents RLVR-World, a framework that applies reinforcement learning with verifiable rewards (RLVR) to optimize world models across different modalities, including language and video. It investigates the use of RLVR for fine-tuning language models in text games and web navigation tasks, as well as video models for robot manipulation trajectory prediction. The results show performance improvements in prediction accuracy and visual metrics, and the framework is also applied to downstream tasks such as model predictive control and policy evaluation.

**Questions:**

1. " Why does the model's accuracy improve by +34.7% for 'unchanged' cases and by +8.9% for 'changed' cases in Text Game State Prediction?   Does this difference reflect an inherent mechanism within the model for dealing with different situation?"

**Ethical Concerns:**

["NO or VERY MINOR ethics concerns only"]

**Final Justification:**

My main criticism was the lack of SOTA comparisons. The authors gave a sufficient reaspn and completely rectified this with extensive and convincing new experiments. This new evidence fundamentally elevates the paper's contribution.

**Limitations:**

yes

**Quality:**

2

**Strengths And Weaknesses:**

Strength：
1. The idea of using RLVR to optimize world models is novel and represents a significant departure from traditional training objectives.
2. The paper pioneers the application of RLVR in video world models, demonstrating its potential beyond language models. This opens up new avenues for research in generative models for visual data.

Weakness:
1.  The paper does not provide a detailed comparison with other recent advances in video world models

---

> ### Author Rebuttal · Authors · 2025-07-31
>
> We sincerely thank Reviewer ndrJ for the thorough review and valuable questions. We hope our following response can resolve your concerns and strengthen your confidence in the value of our work.
>
> ### Q1: Comparison with Advanced Video World Models
>
> First of all, we would like to clarify that our work does not aim to introduce a new state-of-the-art world model. Instead, we reveal a new training paradigm (RLVR) that can improve existing world models more effectively than traditional training approaches. This paradigm, along with the underlying insight, is general and applicable to a wide range of models. For demonstration, our paper applies it to the recent state-of-the-art world model, iVideoGPT (NeurIPS 2024).
>
> Nevertheless, in response to Reviewer BbTo's suggestion, we have added **comparisons with a number of advanced video world models across three new datasets and two tasks**: next-frame prediction and model predictive control.
>
> Please refer to our response to Q2 of Reviewer BbTo for full experimental details and extended results. Key results are presented as follows.
>
> **Next-Frame Prediction**
>
> We have conducted experiments on three additional datasets (PushT, Rope, Granular), comparing against advanced recurrent latent space models, including DINO-WM (ICML 2025) [1], as well as an action-conditioned diffusion model, AVDC (ICLR 2024) [2]. The results are presented below.
>
> |Method|LPIPS ↓|LPIPS ↓|LPIPS ↓|SSIM ↑|SSIM ↑|SSIM ↑|
> |-|-|-|-|-|-|-|
> | |PushT|Rope|Granular|PushT|Rope|Granular|
> |R3M|0.045|0.023|0.080|0.956|0.982|0.917|
> |ResNet|0.063|0.025|0.080|0.950|0.980|0.915|
> |DINO CLS|0.039|0.029|0.086|0.973|0.980|0.912|
> |AVDC|0.046|0.060|0.106|0.959|0.979|0.909|
> |DINO-WM (Reported)|**0.007**|**0.009**|0.035|**0.985**|**0.985**|0.940|
> |DINO-WM (Public checkpoint)|0.0339|-|-|0.9638|-|-|
> |Base (Ours)|0.0083|0.0303|0.0314|0.9828|0.9786|0.9479|
> |RLVR-World (Ours)|**0.0070**|0.0208|**0.0242**|**0.9846**|0.9814|**0.9542**|
>
> Overall, our model, after applying RLVR, achieves performance comparable to the strongest baseline, DINO-WM, and significantly outperforms all other baselines. Notably, it surpasses DINO-WM by a substantial margin on the most challenging particle-based dataset, Granular.
>
> **Model Predictive Control (MPC)**
>
> We also conduct MPC experiments, comparing with state-of-the-art world models, including DreamerV3 (Nature 2025), TD-MPC2 (ICLR 2024), IRIS (ICLR 2023), and DINO-WM.
>
> | Model              | IRIS | DreamerV3 | TD-MPC2 | DINO-WM  | Base (Ours) | RLVR-World (Ours) |
> | ------------------ | ---- | --------- |-|-| ----------- | ----------------- |
> | PushT Success Rate | 0.32 | 0.30|0.00| **0.86** | 0.80        | **0.86**          |
>
> The results show that RLVR also improves the control performance of our base model, achieving the strongest results and outperforming popular world model methods such as DreamerV3 and TD-MPC2.
>
> ### Q2: Discussion on Text Game State Prediction Accuracy
>
> For the accuracy difference in unchanged and changed cases:
>
> 1. **The difference in accuracy and RL gains stems from differences in task difficulty**. Changed cases are inherently more challenging than Unchanged ones: for Unchanged cases, the model only needs to correctly predict that no objects have changed; for Changed cases, the model must not only identify which objects will change but also accurately predict the outcomes of those changes. Therefore, Changed cases are harder to answer, and improvements are more difficult to achieve.
> 2. Improvement in both cases indicates the model can handle opposing tasks. Unchanged cases favor more conservative prediction ("no changes" is sufficient), while Changed cases require more aggressive predictions to capture object changes. Our approach avoids collapsing into a single scenario, and instead can accommodate both types of tasks and truly enhance the learned relationship between actions and changes.
>
> In summary, these results show that RLVR is applicable to complex and diverse world model settings, **improving performance across multiple task types**, with more substantial gains observed in the simpler task.
>
> [1] DINO-WM: World Models on Pre-trained Visual Features enable Zero-shot Planning. ICML 2025.
>
> [2] Learning to Act from Actionless Videos through Dense Correspondences. ICLR 2024.

---

> > ### Author Response · Authors · 2025-08-06
> > **Waiting eagerly for your response**
> >
> > Dear Reviewer ndrJ,
> >
> > Thank you again for your comprehensive review and valuable feedback.
> >
> > As we are now more than halfway through the reviewer-author discussion period, we would like to express our sincere appreciation and briefly highlight how we have addressed your concerns:
> >
> > - We **added comparisons with a number of advanced video world models** (including recurrent latent space models like DINO-WM and the diffusion model AVDC) across **three new datasets** (PushT, Rope, Granular) and **two task settings** (next-frame prediction and model predictive control).
> > - We elaborate on our insight into the accuracy difference between the two cases in the text game state prediction experiment.
> >
> > Please let us know if there are any remaining concerns not addressed in our rebuttal. We would be more than happy to clarify and engage in further discussions.
> >
> > If all concerns have been resolved, we would appreciate if you would consider re-evaluating your assessment of our work.
> >
> > Sincerely,
> >
> > Authors

---

> > ### Author Response · Authors · 2025-08-08
> > **Discussion period ends soon**
> >
> > Dear Reviewer ndrJ,
> >
> > We would like to thank all reviewers for their reviews. We are encouraged that our work has been positively evaluated by Reviewer NuUF and d8tC through constructive discussions.
> >
> > As the **discussion period ends tomorrow**, we would greatly value **your response to our rebuttal**, so that we can hear your feedback and address any remaining questions promptly. For your convenience, we have provided a brief summary of our rebuttal in our previous message.
> >
> > If your concerns have been addressed, we would sincerely appreciate it if you could consider raising your score.
> >
> > Sincerely,
> >
> > Authors.

---

> ### Comment · Area_Chair_mVRE · 2025-08-06
>
> Dear ndrJ, we need you to show up and engage in the discussion phase.
>
> The guideline is:
>
> - read the author rebuttal
> - engage in discussions (reviewers must talk to authors, and optionally to other reviewers and AC - ask questions, listen to answers, and respond to authors)
> - fill in "Final Justification" text box and update “Rating” accordingly (this can be done upon convergence - reviewer must communicate with authors first)
>
> The (new) deadline is Aug 8, 11.59pm AoE.

---

> ### Comment · Reviewer_ndrJ · 2025-08-08
> **Thanks for the response**
>
> Thank you to the authors for the detailed and thorough rebuttal. The new experiments comparing RLVR-World with strong recent baselines have fully addressed my primary concern regarding the lack of comparative evaluation. And your explanation for the accuracy difference in the text game state prediction task is clear. I'm rasing my score accordingly.

---

> > ### Author Response · Authors · 2025-08-09
> > **Appreciation for your response and positive evaluation**
> >
> > Dear Reviewer ndrJ,
> >
> > We are very pleased to hear that your concerns have been fully addressed. Your review has been invaluable in helping us strengthen our work, and we will incorporate the new experimental results into the revised main paper.
> >
> > Thank you for taking the time to review our work and for the final positive evaluation.
> >
> > Sincerely,
> >
> > Authors.

---

### Official Review · Reviewer_BbTo · 2025-07-03

**Clarity:** 3
**Significance:** 2
**Originality:** 2
**Rating:** 4
**Confidence:** 4

**Summary:**

This paper proposes to train world model on text or video based transition datasets with reinforcement learning instead of the conventional maximum likelihood estimation. Specifically, the inputs of states and actions are tokenized and the model is trained to predict the next state via reinforcement learning. The authors evaluate the proposed method on text-based transition dataset and a manipulation task, and show that the proposed method can achieve better performance than the supervised fine-tuning (SFT) baseline.

**Questions:**

See strengths and weaknesses part.

**Ethical Concerns:**

["NO or VERY MINOR ethics concerns only"]

**Final Justification:**

Additional experiments addressed part of my concerns. While the overall technical novelty remains limited and new experiments on MPC in PushT do not show improved performance over previous work.

**Limitations:**

Yes

**Paper Formatting Concerns:**

No.

**Quality:**

3

**Strengths And Weaknesses:**

## Strengths

* The paper is well-written and clearly presents the motivation, method, and results of the proposed approach. The clarity of the writing makes it easy to follow the logic and understand the contributions.

## Weaknesses

* The overall technical contribution of the paper is limited. The tokenization, model architecture, and training method are not novel. This paper does not provide a significant insight or advancement in this field.

* The proposed method of training a world model is very general, while the evaluations are mostly limited to either text form under a language model or a single visual dataset of manipulation. There are diverse datasets in the field of world model, and experiments in other popular domains of robotic or gaming simulations are worthwhile to conduct [r1].

* In experiments, there are very few baselines for comparison with the proposed world model training. The main comparisons are executed based on vanilla pre-trained models and SFT. Many other methods of world model like recurrent latent state models are also important baselines [r1].

* The size of the language model used is much smaller than the state-of-the-art models, which may limit the scalability of the proposed method over larger models.

* It seems that there is no experiments over model predictive control (MPC) in the manipulation task. One major application of world models is to enable MPC in robotic manipulation tasks without interaction with the true environment. The paper would benefit from including experiments on MPC with the proposed world model.

[r1] Zhou G, Pan H, LeCun Y, Pinto L. Dino-wm: World models on pre-trained visual features enable zero-shot planning. arXiv preprint arXiv:2411.04983. 2024 Nov 7.

---

> ### Author Rebuttal · Authors · 2025-07-31
>
> We sincerely thank Reviewer BbTo for the valuable and comprehensive feedback. We have made our best effort to provide additional clarifications and experimental results to address your concerns.
>
> ### Q1: Novelty and Insights
>
> We respectfully disagree with the comment that our work lacks significant insight or advancement in this field, and we apologize for not articulating our contributions clearly enough.
>
> As illustrated in Figure 1 and the Introduction, world models are becoming increasingly capable but also more complicated, often involving non-end-to-end architectures and training objectives. To our knowledge, our work is **the first to identify the following key insight**:
>
> > As world models are built with more advanced models, their training methods diverge further from the actual task objective of world modeling.
>
> This fundamental observation has not been discovered or addressed in prior work on autoregressive world models [1,2] or diffusion-based world models.
>
> This necessitates new and more aligned training methods, especially after large-scale pre-training with traditional surrogate objectives. In this context, **identifying RLVR as a direct and effective training method to meet the above insight, is a novel and meaningful contribution**. Fully leveraging this insight leads to substantial improvements: as shown in Figure 3, the gains achieved in just a few hundred RL steps cannot even be achieved by 1000x more steps of standard next-token prediction training. We also consider **this first successful application of RLVR to video world models** as a minor but still novel technical contribution.
>
> Although our work may not introduce entirely new techniques, we believe this paradigm-shifting insight is both valuable and impactful. We are encouraged that other reviewers recognized this contribution—for instance, Reviewer ndrj described the idea as "novel and a significant departure from traditional training objectives," while Reviewer d8tC noted that we "broaden the applicability" of RLVR.
>
> ### Q2: Additional Datasets & Baselines
>
> To provide a more comprehensive evaluation, we include additional datasets and advanced baselines, following the reviewer's suggestion.
>
> **Datasets**: We adopt **three datasets (Push-T, Rope, and Granular)**, from DINO-WM [3]. We strictly follow the evaluation protocol described in the original paper and official code, including image resolution, training-validation split, number of history frames, and frame skip.
>
> **Baselines**: We compare our models against all recurrent latent state models provided in [3], which use different latent spaces from pretrained encoders such as R3M, ResNet, and DINO. We also include an action-conditioned diffusion model, AVDC.
>
> **Results**: The learning curves of our models on three datasets are shown below (Due to NeurIPS policy, we cannot provide figures in rebuttal).
>
> |Training step|100000|200000|300000 (Pretrain)|(RLVR) 60|100|200|300|430|
> |-|-|-|-|-|-|-|-|-|
> |PushT LPIPS|0.0119|0.0090|0.0083|0.0078|0.0075|0.0074|0.0072|**0.0070**|
>
> |Training step|20000|40000|60000|70000 (Pretrain)|(RLVR) 100|200|400|720|
> |-|-|-|-|-|-|-|-|-|
> |Rope LPIPS|0.0373|0.0323|0.0313|0.0303|0.0260|0.0249|0.0225|**0.0208**|
>
> |Training step|10000|20000|30000 (Pretrain)|(RLVR) 40|100|200|300|500|
> |-|-|-|-|-|-|-|-|-|
> |Granular LPIPS|0.0382|0.0322|0.0314|0.0268|0.0255|0.0256|0.0246|**0.0242**|
>
> Across all datasets, **RLVR consistently improves upon our base model**. Notably, when training with next-token prediction on smaller datasets (Rope and Granular, each containing only 1,000 episodes of length 20), the models are prone to overfitting and require early stopping. In contrast, RLVR continues to significantly enhance prediction performance on the validation set, demonstrating greater robustness and generalization.
>
> We then compare the prediction capabilities of different world models. Since DINO-WM only releases model checkpoints for the Push-T dataset among the three, we additionally include our own evaluation results using the publicly available Push-T checkpoint.
>
> |Method|LPIPS ↓|LPIPS ↓|LPIPS ↓|SSIM ↑|SSIM ↑|SSIM ↑|
> |-|-|-|-|-|-|-|
> | |PushT|Rope|Granular|PushT|Rope|Granular|
> |R3M|0.045|0.023|0.080|0.956|0.982|0.917|
> |ResNet|0.063|0.025|0.080|0.950|0.980|0.915|
> |DINO CLS|0.039|0.029|0.086|0.973|0.980|0.912|
> |AVDC|0.046|0.060|0.106|0.959|0.979|0.909|
> |DINO-WM (Reported)|**0.007**|**0.009**|0.035|**0.985**|**0.985**|0.940|
> |DINO-WM (Public checkpoint)|0.0339|-|-|0.9638|-|-|
> |Base (Ours)|0.0083|0.0303|0.0314|0.9828|0.9786|0.9479|
> |RLVR-World (Ours)|**0.0070**|0.0208|**0.0242**|**0.9846**|0.9814|**0.9542**|
>
> **Our model after RLVR achieves overall performance comparable to the strongest baseline**, DINO-WM, and outperforms it by a significant margin on the most challenging particle dataset, Granular.
>
> Our model underperforms DINO-WM on the Rope dataset. We hypothesize that this is due to our model being prone to overfitting on this relatively small dataset. In contrast, recurrent latent state models that leverage large-scale pretrained visual encoders demonstrate greater robustness. To validate this hypothesis, we explore training a base model jointly on the Rope and Granular datasets (both collected from PyFlex simulators), followed by fine-tuning with RLVR on each individual dataset:
>
> |Method|LPIPS ↓|LPIPS ↓|SSIM ↑|SSIM ↑|
> |-|-|-|-|-|
> | |Rope|Granular|Rope|Granular|
> |Base (Ours, jointly trained)|0.0259|0.0341|0.9795|0.9454|
> |RLVR-World (Ours, individually fine-tuned)|**0.0165**|**0.0244**|**0.9829**|**0.9540**|
>
> We observe that the LPIPS on Rope improves from 0.0208 to 0.0165. Looking forward, we believe the potential of RLVR can be further unlocked when applied to large-scale pre-trained world models, mirroring the remarkable success of LLMs.
>
> ### Q3: Scaling Language Models
>
> We appreciate the reviewers' rigor and agree that, in the modern era, the effectiveness of LLM algorithms should ultimately be validated in truly large-scale settings. At the same time, we believe that scientific progress is inherently incremental. The foundational algorithm behind RLVR, GRPO, was initially validated on a relatively small-scale language model, DeepSeekMath 7B. We believe our current results remain valuable and can inspire further exploration within the community.
>
> Nonetheless, we have made our best effort to scale our text game state prediction experiments to 7B-parameter language models. Specifically, we use DeepSeek-R1-Distill-Qwen-7B as our base model.
>
> |Model|Unchanged Acc.|Changed Acc.|Overall Acc.|
> |-|-|-|-|
> |Base (1.5B)|11.98%|0.08%|7.11%|
> |SFT (1.5B)|38.88%|24.21%|32.87%|
> |RLVR-World (1.5B, Ours)|73.57% (+34.69%)|33.14% (+8.93%)|57.01% (+24.14%)|
> |Base (7B)|46.90%|5.53%|29.92%|
> |SFT (7B)|65.94%|31.32%|51.76%|
> |RLVR-World (7B, Ours)|**83.08%** (+17.14%)|40.33% (+9.01%)|**65.53%** (+13.77%)|
> |GPT-4|73.90%|**51.60%**|64.76%|
>
> The results demonstrate that the effectiveness of RLVR-World scales to 7B models, with our final overall accuracy matching that of GPT-4. We hope these new results further strengthen your confidence in the scalability of our method.
>
> ### Q4: Model Predictive Control
>
> We first note that our original submission already includes experiments on policy evaluation, a critical subtask of model predictive control (MPC). Conducted in the realistic RT-1 domain, this experiment is both challenging and well-suited to demonstrate the benefits of our method for robotic manipulation.
>
> Due to time constraints, it is not feasible for us to set up real robot control experiments. Instead, we conduct additional MPC experiments in the simulated Push-T environment, following the setup used in DINO-WM.
>
> **Setup**: We strictly follow the same planner configuration as DINO-WM, including goal construction, number of samples, and optimization steps. Since our model predicts discrete tokens and raw pixels without a compact latent space, we use the publicly available DINOv2 encoder to embed predicted frames and compare them to the goal observation during planning. We find that the DINO latent space is more effective for planning than pixel-level MSE or LPIPS, consistent with the findings in DINO-WM. We emphasize that our MPC experiments are intended to compare world models, and the choice of distance metric is orthogonal to our main contributions.
>
> We do not conduct experiments on Rope and Granular due to the lack of MPC configurations and details in the official DINO-WM code, making it difficult to replicate the setup.
>
> **Results**: The results below show that RLVR also enhances the control performance of our base model, achieving results comparable to DINO-WM.
>
> |Model|IRIS|DreamerV3|TD-MPC2|CEM with DINO-WM|CEM with Our Base|CEM with Our RLVR-World|
> |-|-|-|-|-|-|-|
> |PushT Success Rate|0.32|0.30|0.00|**0.86**|0.80|**0.86**|
>
> ### In Summary
>
> We sincerely thank the reviewer once again, especially for pointing us to the relevant work, DINO-WM. The additional experiments on **three new datasets and two task setups, with comparisons against several advanced world models**, further strengthen our contributions. We will include all experimental settings, quantitative results, and visualizations of both prediction and planning in the next version of our paper.
>
> [1] Transformers are Sample-Efficient World Models. ICLR 2023.
>
> [2] iVideoGPT: Interactive VideoGPTs are Scalable World Models. NeurIPS 2024.
>
> [3] DINO-WM: World Models on Pre-trained Visual Features enable Zero-shot Planning. ICML 2025.

---

> > ### Author Response · Authors · 2025-08-06
> > **Waiting eagerly for your response**
> >
> > Dear Reviewer BbTo,
> >
> > Thank you again for your comprehensive review and valuable feedback.
> >
> > As we are now more than halfway through the reviewer-author discussion period, we would like to express our sincere appreciation and briefly highlight how we have addressed your concerns:
> >
> > - We clarified our core insight and contribution: **identifying RLVR as a more task-aligned and effective post-training paradigm for world models, and successfully applies it for the first time to both language and video world models**. We are encouraged that this contribution has been positively acknowledged by Reviewer NuUF, who described it as “meaningful and publication-worthy.”
> > - Following your suggestion, we conducted **additional evaluations on three new datasets** (PushT, Rope, Granular), and **compared against advanced world model baselines** (including recurrent latent space models like DINO-WM and the diffusion model AVDC).
> > - We also performed a model-predictive control (MPC) experiment on the PushT environment to further validate the effectiveness of our method.
> > - We scaled our language model experiments to 7B parameters, further demonstrating the scalability of our method.
> >
> > Please let us know if there are any remaining concerns not addressed in our rebuttal. We would be more than happy to clarify and engage in further discussions.
> >
> > If all concerns have been resolved, we would appreciate if you would consider re-evaluating your assessment of our work.
> >
> > Sincerely,
> >
> > Authors

---

> > ### Author Response · Authors · 2025-08-08
> > **Discussion period ends soon**
> >
> > Dear Reviewer BbTo,
> >
> > We would like to thank all reviewers for their reviews. We are encouraged that our work has been positively evaluated by Reviewer NuUF and d8tC through constructive discussions.
> >
> > As the **discussion period ends tomorrow**, we would greatly value **your response to our rebuttal**, so that we can hear your feedback and address any remaining questions promptly. For your convenience, we have provided a brief summary of our rebuttal in our previous message.
> >
> > If your concerns have been addressed, we would sincerely appreciate it if you could consider raising your score.
> >
> > Sincerely,
> >
> > Authors.

---

> > > ### Comment · Reviewer_BbTo · 2025-08-08
> > > **Thanks for the response**
> > >
> > > I would like to thank the authors for their detailed explanations and the additional experiments. The response has fully addressed my concerns over evaluating datasets, baselines comparison and model size. I appreciate the significant effort the authors put into the extra experiments and analyses, and I have accordingly increased my score to 4.
> > >
> > > However, my concern regarding the limited technical novelty remains as the main technique developed in this work are existing well-established techniques. As such, the primary contribution is more on empirical evaluations rather than methodology development/innovation.
> > >
> > > Additionally, I am still confused about the authors’ claim concerning policy evaluation experiments on RT-1 in the context of MPC. It seems that the experiments here in the original submission are on policy evaluation using world models for rollout, rather than actual planning/control towards a particular goal and planning algorithms like CEM. New results on PushT here also indicate that the performance is comparable to that of DINO-WM.

---

> > > > ### Author Response · Authors · 2025-08-09
> > > > **Appreciation for you response and positive evaluation**
> > > >
> > > > Dear Reviewer BbTo,
> > > >
> > > > We sincerely thank you for the time you have invested in carefully reading our rebuttal, examining the new experiments, and providing detailed feedback. Your thoughtful follow-up and positive evaluation greatly encourage us.
> > > >
> > > > We would like to respectfully respond to your latest comments:
> > > >
> > > > - **Technical novelty**: We fully respect that different researchers may have different academic perspectives on what constitutes a contribution to the field. In our humble opinion, our work opens up a new way to address the pressing challenge in this rapidly evolving field, despite using existing well-established techniques. We will more accurately refine our claims to emphasize the paradigm-shifting aspect of our work, rather than technical novelty.  We will also explicitly discuss this limitation and highlight the development of novel RLVR methods for world modeling as an important direction for future research.
> > > > - **Policy evaluation**: In our previous response, we regard policy evaluation as a *subtask* of MPC, since a key step in MPC is to evaluate the optimality of actions from different distributions, based on which action selection can be improved. We acknowledge that this viewpoint may not be consistently accepted by the community, and it does not replace the need to validate our method in real robot control. Given that setting up real-world experiments requires substantial engineering resources and is beyond the current scope of this paper, we will discuss this as a critical future work.
> > > > - **MPC performance on PushT**: We consider the comparable performance to DINO-WM reasonable, as this visually simple environment allows both DINO-WM and our method to achieve sufficiently high and similar visual fidelity. The current control performance may be limited by other factors like planning algorithms, rather than by differences in world model quality. We emphasize that all our experiments are designed primarily to validate the *effectiveness of RLVR for world models*, and matching state-of-the-art performance is just a side effect. We also believe that RLVR could be applied to broader models like DINO-WM, particularly given concurrent developments of GRPO algorithms for deterministic models such as flow matching [1].
> > > >
> > > > Once again, we are very grateful for your time, effort, and constructive feedback. Your comments have directly led to significant improvements in our work, including substantial new experimental evaluations.
> > > >
> > > > Sincerely,
> > > >
> > > > Authors
> > > >
> > > > [1] DanceGRPO: Unleashing GRPO on Visual Generation. arXiv 2025.05.

---

> ### Comment · Area_Chair_mVRE · 2025-08-06
>
> Dear BbTo, we need you to show up and engage in the discussion phase.
>
> The guideline is:
> - read the author rebuttal
> - engage in discussions (reviewers must talk to authors, and optionally to other reviewers and AC - ask questions, listen to answers, and respond to authors)
> - fill in "Final Justification" text box and update “Rating” accordingly (this can be done upon convergence - reviewer must communicate with authors first)
>
> The deadline is Aug 8, 11.59pm AoE.

---

### Decision · Program_Chairs · 2025-09-17

**Decision:**

Accept (poster)

**Comment:**

This paper proposes to train world models for downstream task objectives, by tokenizing the states and actions and asking the model to predict next state via reinforcement learning, instead of the conventional MLE approach. This idea has been explored in LLM research but not in robotics/worldmodels until this work. The results are strong in multiple domains. The paper received a consensus of positive reviews: 4, 4, 5, 5. Initially the reviewers pointed out that the paper was missing some SOTA comparisons, but sufficiently convincing experiments were added in the rebuttal to turn all reviewers positive. The AC sides with the consensus and recommends acceptance.